# How Particle System Theory Enhances Hypergraph Message Passing

**Yixuan Ma**
Shanghai Jiao Tong University
mayx5901@sjtu.edu.cn

**Kai Yi**
University of Cambridge
ky347@cam.ac.uk

**Pietro Liò**
University of Cambridge
pl219@cam.ac.uk

**Shi Jin**
Shanghai Jiao Tong University
shijin-m@sjtu.edu.cn

**Yu Guang Wang**\*
Shanghai Jiao Tong University
yuguang.wang@sjtu.edu.cn

## Abstract

Hypergraphs effectively model higher-order relationships in natural phenomena, capturing complex interactions beyond pairwise connections. We introduce a novel hypergraph message passing framework inspired by interacting particle systems, where hyperedges act as fields inducing shared node dynamics. By incorporating attraction, repulsion, and Allen-Cahn forcing terms, particles of varying classes and features achieve class-dependent equilibrium, enabling separability through the particle-driven message passing. We investigate both first-order and second-order particle system equations for modeling these dynamics, which mitigate over-smoothing and heterophily thus can capture complete interactions. The more stable second-order system permits deeper message passing. Furthermore, we enhance deterministic message passing with stochastic element to account for interaction uncertainties. We prove theoretically that our approach mitigates over-smoothing by maintaining a positive lower bound on the hypergraph Dirichlet energy during propagation and thus to enable hypergraph message passing to go deep. Empirically, our models demonstrate competitive performance on diverse real-world hypergraph node classification tasks, excelling on both homophilic and heterophilic datasets. Source code is available at the link.

## 1 Introduction

Hypergraph Neural Networks (HNNs) [20], built upon Graph Neural Networks (GNNs) [29], have demonstrated remarkable success in modeling higher-order relationships involving multiple nodes [28]. In complex systems such as social networks [5, 32, 50] and biomolecular interactions [41, 46], hypergraphs can effectively capture complex group dynamics compared to pairwise graphs.

The multi-node, higher-order nature of hypergraphs naturally lends itself to an analogy with particle dynamical systems. This is because hypergraph message passing, much like the inherent interactions in particle motion, fundamentally involves multiple interacting components. Driven by this insight, we introduce Hypergraph Atomic Message Passing (HAMP), a novel framework that reframes hypergraph message passing through the lens of interacting particle systems. Our key innovation in HAMP is the conceptualization of each hyperedge as a dynamic field that governs the shared dynamics of nodes. As Figure 1 shows, HAMP updates hypergraph embeddings by superimposing the forces exerted by the hypergraph's nodes (particles). This framework offers significant advantages, including

---

\*Corresponding author

39th Conference on Neural Information Processing Systems (NeurIPS 2025).

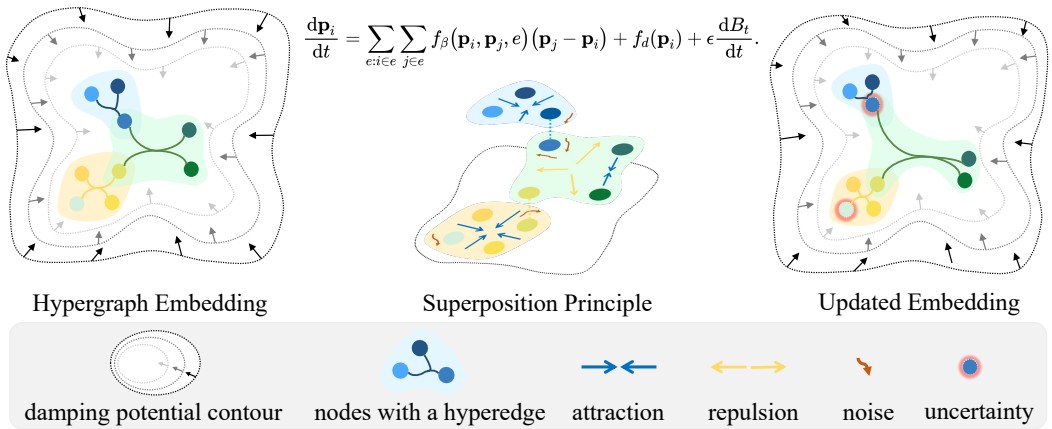

$$\frac{d\mathbf{p}_i}{dt} = \sum_{e:i\in e}\sum_{j\in e} f_\beta(\mathbf{p}_i, \mathbf{p}_j, e)(\mathbf{p}_j - \mathbf{p}_i) + f_d(\mathbf{p}_i) + \epsilon\frac{dB_t}{dt}.$$

| Hypergraph Embedding | Superposition Principle | Updated Embedding |

| damping potential contour | nodes with a hyperedge | attraction | repulsion | noise | uncertainty |

Figure 1: An illustration for HAMP framework. The property $\mathbf{p}$ can account for feature or velocity.

the ability to mitigate over-smoothing and facilitate deep message passing on both homophilic and heterophilic datasets, a claim supported by our experimental findings in Section 6.3.

Particle theory takes various forms, traditionally categorized into first-order and second-order systems. While first-order systems offer a direct approach to defining evolution, the second-order systems are inherently more stable, converging towards an asymptotically stable state. Both first-order and second-order systems exhibit desirable separability properties stemming from the balance between attractive and repulsive forces [27, 31]. Repulsive forces are crucial for separating distinct features, while attractive forces encourage particles of the same category to cluster, allowing for the refinement of essential features. Through this mechanism, we construct the interaction forces within HAMP.

Inspired by particle systems and informed by experimental observations, we identify a critical need for a delicate balance between attractive and repulsive forces. Excessive attraction often leads to feature over-smoothing [34, 35], while unchecked repulsion can cause feature explosion [44]. To address these issues and ensure system stability, we introduce the Allen-Cahn force [1] as a balancing mechanism. We theoretically demonstrate that the resulting system exhibits favorable separability and that its solutions converge to an equilibrium state, thereby mitigating both over-smoothing and feature explosion. Our focus on second-order systems is further motivated by their inherent theoretical stability, which ensures robustness even with deeper network layers. In practical tasks, ambiguous cases where distinct states are hard to differentiate often lead to uncertainty [42]. To address this, we incorporate a stochastic term into HAMP, driven by Brownian motion, which results in a stochastic differential equation. Our main contributions are summarized as follows:

- We propose HAMP, a novel hypergraph message passing framework based on particle system theory, filling in the theoretical gap in understanding hypergraph message passing from the perspective of particle system.

- We design two hypergraph message passing algorithms, HAMP-I and HAMP-II, constructed through first-order and second-order particle dynamical systems.

- Theoretically, we prove that HAMP maintains a strictly positive lower bound on the hypergraph Dirichlet energy, effectively resisting over-smoothing.

- Empirically, through numerical experiments, we should that HAMP achieves competitive results on node classification benchmarks. Notably, on heterophilic hypergraphs, HAMP consistently outperforms the current state-of-the-art baselines by a margin of 1–3%.

## 2 Preliminaries

**Hypergraphs.** A hypergraph is a generalization of a graph in which an edge can join any number of vertices. In contrast, in an ordinary graph, an edge connects exactly two vertices. We can denote a hypergraph as $\mathcal{G} = \{\mathcal{V}, \mathcal{E}\}$, with $|\mathcal{V}|$ nodes and $|\mathcal{E}|$ hyperedges, where $\mathcal{V} = \{v_1, \ldots, v_{|\mathcal{V}|}\}$ is the set of nodes and $\mathcal{E} = \{e_1, \ldots, e_{|\mathcal{E}|}\}$ is the set of hyperedges. $\mathcal{E}(i)$ denotes a set containing all the nodes

sharing at least one hyperedge with node $i$. The incidence matrix $\mathbf{H} \in \mathbb{R}^{|\mathcal{V}| \times |\mathcal{E}|}$ is a common notation in hypergraph. Its entries are defined as $\mathbf{H}_{i,e} = 1$ if node $i$ belongs to hyperedge $e$, and 0 otherwise.

**Message Passing in Hypergraphs.** Neural message passing [23, 3] is the most widely used propagator for node feature updates in GNNs, which propagates node features while taking into account their neighboring nodes. Next, we generalize graph message passing to hypergraph, where multiple nodes interaction reflected in a hyperedge is considered,

$$
\mathbf{x}_i^{(l+1)} = \Psi^{(l)} \left( \mathbf{x}_i^{(l)}, \Phi_2 \left( \{ \mathbf{z}_e^{(l)}, \Phi_1(\{\mathbf{x}_j^{(l)}\}_{j \in e}) \}_{e \in \mathcal{E}(i)} \right) \right), \tag{1}
$$

where $\mathbf{z}_e^{(l)}$ denotes the feature embedding of hyperedge in the $l^{th}$ layer. $\Phi_1^{(l)}$ and $\Phi_2^{(l)}$ denote differentiable permutation invariant functions (e.g., sum, mean, or max) for nodes and hyperedges, respectively. $\Psi^{(l)}$ denotes a differentiable function such as MLPs.

**Neural ODEs.** Neural ODEs [18, 10] are essentially ordinary differential equations (ODEs), where the time derivative of the hidden states are parameterized by a neural network $f_\theta$,

$$
\frac{\mathrm{d}\mathbf{x}(t)}{\mathrm{d}t} = f_\theta\left(\mathbf{x}(t), t\right), \quad \mathbf{x}(0) = \mathbf{x}, \tag{2}
$$

where $t$ denotes time, $\mathbf{x}(t)$ are the system state, and $\theta$ is learnable parameter. It can be understood as the continuous limit form of the residual network, e.g, $\mathbf{x}(t + \tau) = f_\theta\left(\mathbf{x}(t), t\right) + \mathbf{x}(t)$, where $\tau$ denotes the step size of time. Thus, Neural ODEs model the continuous dynamics system that evolve hidden states over a continuous range of "depths", analogous to layers in traditional deep networks.

## 3 Hypergraph Message Passing based on Particle Dynamics

Viewing HNNs as continuous systems allows for the application of physical models to elucidate and investigate their properties. A compelling conceptual analogy arises when comparing hypergraph message passing in deep learning with interacting particle systems studied in statistical physics. These disparate fields display remarkable structural and dynamical resemblances: the local potential field-governed interactions among particles in many-body systems are analogous to the aggregation operations in HNNs. Concurrently, the emergent collective behaviors observed in self-organizing dynamics demonstrate a functional equivalence to information propagation mechanisms in HNNs. As message passing delineates the transfer of information between entities, particle systems can represent this by having particles acting as entities that exchange information through the transition of specific attributes. Moreover, such particle systems can be endowed with hypergraph structures by interpreting hyperedges as distinct fields that exert influence over all nodes encompassed by them.

**Fields and Hypergraphs.** The hypergraph uses hyperedges which reflect the common relationship among multiple nodes. Such multi-nodal relationships are aptly described using "fields". Each field imposes a common dynamic on the nodes under its influence, which is equivalent to a hyperedge defining a unified information propagation step for its constituent nodes. A node participating in multiple hyperedges thus receives information from diverse sources due to field superposition. Thus, hyperedges reflect both the interactions of particles within a specific region and the concurrent influence of the external environment, which together dictate the propagation dynamics. Based on these observations, we propose a hypergraph message passing framework for describing information dynamics. We formalize a composite field for each node $i$ by $F_i = \sum_{e:i \in e} F_i^e + F_d$ to encapsulate the hypergraph structure. This field $F_i$ aggregates influences akin to those in a particle system, where $F_i^e$ represents the interaction energy for node $i$ with the edge $e$, and $F_d$ is the damping energy.

**Interaction Force.** Interaction force forms a basic association between hypergraph message passing and particle dynamics. We consider particles identified by their specific attributes $\mathbf{p}_i \in \mathbb{R}^d$ in the particle system. Then, we define the interaction energy as following

$$
\sum_{e:i \in e} F_i^e := \frac{1}{2} \sum_{e:i \in e} \sum_{j \in e} (\mathbf{p}_i - \mathbf{p}_j)^\top f_\beta(\mathbf{p}_i, \mathbf{p}_j, \mathbf{e})(\mathbf{p}_i - \mathbf{p}_j), \quad e \in \mathcal{E}. \tag{3}
$$

If $f_\beta(\mathbf{p}_i, \mathbf{p}_j, e) > 0$, node $i$ is attracted by node $j$. If $f_\beta(\mathbf{p}_i, \mathbf{p}_j, e) < 0$, node $i$ is repulsed by node $j$. They correspond to two basic interaction forces: attraction and repulsion. Note that in the previous model similar to Eq. 3, the only interaction is attraction. It relies on an implicit belief that all the hyperedges only connect similar nodes. But this is not always true in the real world. Hence, introducing repulsion into hypergraph neural networks is significant.

**Damping Force.** The specific formulation of the damping term can vary. The inclusion of damping in neural networks serves two principal purposes. Firstly, it is crucial for ensuring system stability. If inter-particle repulsive forces do not attenuate adequately with increasing distance, particles risk unbounded separation or unbounded Dirichlet energy, which could lead to network collapse. A damping term mitigates this by exerting a dominant influence on the dynamics, especially near the system's boundaries. Secondly, the damping term has been demonstrated to promote various collective dynamics in numerous systems [30, 31]. For damping, we take double-well potential $F_d = \zeta(1 - \mathbf{p}^2)^2$ with the coefficient $\zeta$. The corresponding damping force is given by the Allen-Cahn force $f_d(\mathbf{p}) = \nabla_v F_d = \delta(1 - \mathbf{p}^2)\mathbf{p}$ [38, 19], where $\delta$ denotes the strength coefficient. This formulation exhibits favorable separation properties, as will be demonstrated in Section 5.

**Hypergraph Dynamics.** Interaction term can be formulated using two primary approaches: reductive and non-reductive expressions. Reductive methods, such as those based on convolution coefficients [22] or attention coefficients [2], often simplify hyperedges into clique-like structures, which can lead to the loss of crucial high-order information. In contrast, non-reductive expressions, like star expansion, aim to preserve this high-order information by explicitly connecting each node to the hyperedges it belongs to. We take the non-reductive expression and Allen-Cahn force to design interaction force $f_\beta$ and damping force $f_d$. In fact, the particle attribute $\mathbf{p}$ can represent diverse properties, such as features or velocity. This flexibility in defining $\mathbf{p}$ naturally allows for the formulation of both first-order and second-order particle systems.

**First-order ODEs.** Inspired by the opinions dynamics [33], we have $\mathbf{p}_i \mapsto \mathbf{x}_i \in \mathbb{R}^d$ that interact with each other according to the first-order system. Taking gradient of composite field $F$, we have

$$\frac{\mathrm{d}\mathbf{x}_i}{\mathrm{d}t} = \sum_{e:i\in e} \sum_{j\in e} f_\beta(\mathbf{x}_i, \mathbf{x}_j, e)(\mathbf{x}_j - \mathbf{x}_i) + f_d(\mathbf{x}_i), \tag{4}$$

where $f_\beta$ is a parameterized function. This form has connection with diffusion process of hypergraph. We defer the detailed analysis to Section 4.

**Second-order ODEs.** Inspired by the flocking dynamics [33, 15], this is second-order model where the attribute is the velocity, $\mathbf{p}_i \mapsto \mathbf{v}_i \in \mathbb{R}^d$, which is coupled to the feature $\mathbf{x}_i \in \mathbb{R}^d$. In this sense, information can be propagated by the evolution of feature and velocity of nodes,

$$\frac{\mathrm{d}\mathbf{v}_i}{\mathrm{d}t} = \sum_{e:i\in e} \sum_{j\in e} f_\beta(\mathbf{x}_i, \mathbf{x}_j, e)(\mathbf{v}_j - \mathbf{v}_i) + f_d(\mathbf{v}_i), \text{ where } \frac{\mathrm{d}\mathbf{x}_i}{\mathrm{d}t} = \mathbf{v}_i. \tag{5}$$

**From ODEs to SDEs.** To capture incompleteness and uncertainty within hypergraph data, stochasticity is introduced into the particle system. For instance, such uncertainty can arise when particles from different classes exhibit similar features. To model this inherent randomness, these stochastic processes are often described using stochastic differential equations (SDEs). We assume these SDEs are driven by Brownian motion $B_t$:

$$\mathrm{d}\mathbf{p}(t + \tau) = \nabla F(\mathbf{p}(t))\mathrm{d}t + \epsilon \mathrm{d}B_t, \text{ where } \mathbb{E}[\mathrm{d}B_t] = 0 \text{ and } \mathrm{Cov}(\mathrm{d}B_t) = \mathbf{I}_d, \tag{6}$$

where $\mathbf{I}_d$ denotes the $d \times d$ identity matrix. In essence, SDEs describe the changes in system state under infinitesimal time variations. Here, particle states undergo continuous evolution through both deterministic drift term and stochastic diffusion component. It can describe hypergraph evolution of neural message passing in more realistic scenario. According to [15, 24], this model can achieve self-organization in a finite time, which enables the system to adjust its own state in the shortest time.

**Hypergraph Message Passing.** We introduce a unified hypergraph message passing framework, called *Hypergraph Atomic Message Passing*, or HAMP, given in Eq. 6. By instantiating this framework with either a first-order or a second-order system formulation, we obtain two variants: HAMP-I and HAMP-II, respectively. Detailed implementations of both approaches are provided systematically in Appendix A to ensure clarity and maintain manuscript focus.

# 4 Scale Translation of Hypergraph: Diffusion and Particle Dynamics

We examine hypergraph message passing through two complementary lenses—microscopic and macroscopic. From the microscopic viewpoint, HAMP is cast as a particle-dynamics system, as previously discussed. From the macroscopic standpoint, message propagation is seen as a diffusion process, which reveals that the particle-based formulation naturally subsumes diffusion-based models and overcomes their intrinsic limitations.

**Hypergraph Diffusion.** Consider node feature space $\Omega = \mathbb{R}^d$ and tangent vector field space $T\Omega = \mathbb{R}^d$. For $\mathbf{x}, \mathbf{y} \in \Omega$, $\mathfrak{x}, \mathfrak{y} \in T\Omega$, and $\mathfrak{x}_{i,j} = -\mathfrak{x}_{j,i}$, we adopt the following inner products:

$$\langle \mathbf{x}, \mathbf{y} \rangle = \sum_{i,j} \mathbf{x}_i \mathbf{y}_j, \quad [\mathfrak{x}, \mathfrak{y}] = \sum_{i>j} \sum_{e \in \mathcal{E}} h_{i,j}^e \, \mathfrak{x}_{i,j} \mathfrak{y}_{i,j}. \tag{7}$$

Here $h_{i,j}^e$ is a tuple related to node $i, j$ and hyperedge $e$ and $h_{i,j}^e = 0$ if $\mathbf{H}_{i,e}\mathbf{H}_{j,e} = 0$. We set $h_{i,j}^e$ to satisfy $\sum_j \sum_{e \in \mathcal{E}} h_{i,j}^e = 1$. For any $\mathfrak{u} \in T\Omega$, by the adjoint relation $[\mathfrak{u}, \nabla \mathbf{x}] = \langle \mathbf{x}, \operatorname{div} \mathfrak{u} \rangle$, where $\nabla \mathbf{x} = \mathbf{x}_j - \mathbf{x}_i$, we can derive $(\operatorname{div}\mathfrak{u})_j = \sum_i \sum_{e \in \mathcal{E}} h_{i,j}^e u_i$. Then, the hypergraph diffusion process is

$$\frac{\mathrm{d}\mathbf{x}_i}{\mathrm{d}t} = \operatorname{div}\nabla \mathbf{x}_i = \sum_j \sum_{e \in \mathcal{E}} h_{i,j}^e (\mathbf{x}_j - \mathbf{x}_i). \tag{8}$$

For simplicity, we rewrite Eq. 8 in matrix form $\frac{d\mathbf{x}}{dt} = -\mathcal{L}\mathbf{x}$, where $\mathcal{L} = \mathbf{I} - (\sum_{e \in \mathcal{E}} h_{i,j}^e)$ is a hypergraph operator. If $\mathcal{L}$ is positive semi-definite, we interpret Eq. 8 as a diffusion-type process of hypergraph. Different parameterizations of $h_{i,j}^e$ then yield distinct diffusion equations. For example, applying a forward Euler discretization to Eq. 8 and setting $(\sum_{e \in \mathcal{E}} h_{i,j}^e) = \mathbf{D}_v^{-\frac{1}{2}} \mathbf{H}\mathbf{W}\mathbf{D}_e^{-1}\mathbf{H}^\top \mathbf{D}_v^{-\frac{1}{2}}$ recovers the simplified HGNN [20] without channel mixing.

**Connecting Particle Dynamics.** Since Eq. 8 coincides with the self-organized dynamics in particle system [33], we reinterpret Eq. 8 not as a standard hypergraph diffusion process, but rather as particle dynamics where $h_{i,j}^e$ denotes the interaction force between nodes $i$ and $j$ under field $e$. In fact, Eq. 8 is a special case of Eq. 4 where only attractive forces are considered in the message propagation. As we show in Section 5, this simplification is atypical in particle systems and leads to over-smoothing.

# 5 Theory of Anti-over-smoothing

For diffusion-type hypergraph networks, we define the *hypergraph Dirichlet energy* of $\mathbf{x} \in \mathbb{R}^{N \times d}$ as $\mathbf{E}(\mathbf{x}) := \sum_{i,j=1}^N \sum_{e \in \mathcal{E}} \mathbf{H}_{i,e}\mathbf{H}_{j,e}\|\mathbf{x}_i - \mathbf{x}_j\|^2 = \operatorname{tr}(\mathbf{x}^\top \mathcal{L}\mathbf{x})$. Furthermore, we will define over-smoothing in hypergraph message passing.

**Definition 5.1.** Let $\mathbf{x}^{(l)}$ denote the hidden features of the $l^{th}$ layer. We define over-smoothing in HNNs as the exponential convergence to zero of the layer-wise Dirichlet energy as a function of $l$, i.e., $\mathbf{E}(\mathbf{x}^{(l)}) \leq C_1 e^{-C_2 l}$, with positive constants $C_1$ and $C_2$.

**Why Do HGNN Cause Over-smoothing?** The normalized hypergraph Laplacian matrix is defined by $\mathcal{L} = \mathbf{I} - \mathbf{P}$, where $\mathbf{P}$ is the propagation matrix derived from the incidence matrix $\mathbf{H}$. In an HGNN, the feature at layer $l$ without activation $\sigma$ evolves according to $\mathbf{x}^{(l)} = \mathbf{P}^{l-1}\mathbf{x}^{(0)}\Theta^{(1)} \cdots \Theta^{(l-1)}$. However, this purely diffusive propagation inevitably induces over-smoothing: one can show $\mathbf{E}(\mathbf{x}^{(l)}) \leq C e^{-\gamma^2 l}$ [9], where $\gamma$ is the smallest non-zero positive eigenvalue of $\mathcal{L}$. This over-smoothing behavior stems from the intrinsic diffusion dynamics. Because $\mathcal{L}$ is symmetric positive semi-definite, repeated application of $\mathbf{P}$ causes node representations to decay exponentially towards a limiting state–which is zero, thereby eroding discriminative power.

**Why Do HAMP Avoid Over-smoothing?** We now derive theoretical guarantees for the collective behavior of models Eq. 4 and Eq. 5. They show that the addition of a repulsive force enforces

a positive lower bound on the Dirichlet energy. Technically, we suppose there exists $\{f_\beta^e\}$ such that $\mathcal{I} = \{1, \cdots, N\}$ can be divided into two disjoint groups with $N_1, N_2$ particles respectively: $f_\beta(h_{i,j}^e) \geq 0$, for $\{i, j\} \in \mathcal{I}_1$ or $\mathcal{I}_2$ and $f_\beta(h_{i,j}^e) \leq 0$, otherwise. We designate $\{x_i^{(1)}\} := \{\mathbf{x}_i | i \in \mathcal{I}_1\}$ and $\{x_j^{(2)}\} := \{\mathbf{x}_j | j \in \mathcal{I}_2\}$. Finally, we impose the symmetry $f_\beta(h_{i,j}^e) = f_\beta(h_{j,i}^e)$, which reflects equal-and-opposite interactions under the same field. Under these conditions, one can show that the Dirichlet energy of our system admits a strictly positive lower bound, as follows. We leave the detailed proofs in Appendix B.

**Proposition 5.2** ($L_2$ separation of HAMP-I). *For Eq. 4, suppose the above assumptions are satisfied. Define the mean value $\bar{\mathbf{x}} := \frac{1}{N} \sum_{i=1}^{N} \mathbf{x}_i$, and the second moments $M_2(\mathbf{x}) := \sum_{i=1}^{N} \mathbf{x}_i^2$. Then for sufficiently large $N_1, N_2$, there exist constants $\lambda_-, \lambda_+$, such that if the initial data satisfies $\lambda(0) := \frac{\widehat{M_2}(0)}{\|\bar{\mathbf{x}}^{(1)}(0) - \bar{\mathbf{x}}^{(2)}(0)\|^2} \leq \lambda_+$, then, there holds that the $L_2$ separation*

$$\lambda(t) := \frac{\widehat{M_2}(t)}{\|\bar{\mathbf{x}}^{(1)}(t) - \bar{\mathbf{x}}^{(2)}(t)\|^2} \leq \lambda_- + (\lambda(0) - \lambda_-)e^{-\mu t}, \tag{9}$$

*with a positive constant $\mu$, where $\widehat{M_2}(t) := M_2(\mathbf{x}^{(1)}(t)) + M_2(\mathbf{x}^{(2)}(t))$.*

**Proposition 5.3** ($L_2$ separation of HAMP-II, [19]). *For Eq. 5, we set $0 < S \leq f_\beta(h_{i,j}^e)$ for $\{i, j\} \in \mathcal{I}_1$ and $0 \leq f_\beta(h_{i,j}^e) \leq D$ otherwise, with $k := \max_i \{|\mathcal{E}(i)|\}$. If the initial $\|\bar{\mathbf{x}}^{(1)}(0) - \bar{\mathbf{x}}^{(2)}(0)\| \gg 1$, and if there exists a positive constant $\eta$ such that*

$$\alpha(S - D)k \min\{N_1, N_2\} \geq \delta + \eta, \tag{10}$$

*Then the system has a bi-cluster flocking.*

**Proposition 5.4** (Lower bound of the Dirichlet energy). *If the hypergraph $\mathcal{H}$ is a connected one, for Eq. 4 with the conditions of Theorem 5.2, or for Eq. 5 with conditions of Theorem 5.1 in [19], there exists a positive lower bound of the Dirichlet energy.*

## 6 Experiments

### 6.1 Experiment Setup

We conduct comprehensive experiments to evaluate the proposed models on node classification task. For more experimental details such as datasets and hyperparameters, please refer to Appendix C.

**Datasets.** Following ED-HNN[43], the real-world hypergraph benchmarking datasets span diverse domains, scales, and heterophiilic levels. They can be divided into two groups based on homophily. The homophilic hypergraphs include academic citation networks (Cora, Citeseer, and Pubmed) and co-authorship networks (Cora-CA and DBLP-CA). The heterophilic hypergraphs cover legislative voting records (Congress, House, and Senate) and retail relationships (Walmart).

**Baselines.** The selected baselines cover two types of hypergraph learning frameworks, comprising both reductive and non-reductive approaches. The reductive methods include HGNN [20], HCHA [2], HNHN [14], HyperGCN [47], and HyperND [37]. The non-reductive methods include UniGCNII [26], AllDeepSets [11], AllSetTransformer [11], ED-HNN [43], and HDS$^{ode}$ [48].

### 6.2 Node Classifications on Hypergraphs

In this section, we evaluate HAMP-I and HAMP-II on nine real-world hypergraph benchmarks for the node classification task. Tab. 1 reports the accuracy on both homophilic and heterophilic datasets. As HDS$^{ode}$ does not provide results on these benchmarks, we reproduce its performance using the official open-source code and perform hyperparameter tuning following the original paper. For other baselines, our results are consistent with those reported by ED-HNN. Overall, our models demonstrate competitive performance across all nine datasets. Notably, the improvement is more pronounced on heterophilic datasets, with the largest accuracy gain of 3% observed on Walmart dataset.

Table 1: Node Classification on standard hypergraph benchmarks. The accuracy (%) is reported with a standard deviation from 10 repetitive runs. (Key: **Best**; Second Best; Third Best.)

| Homophilic | Cora | Citeseer | Pubmed | Cora-CA | DBLP-CA |
|---|---|---|---|---|---|
| HGNN | 79.39±1.36 | 72.45±1.16 | 86.44±0.44 | 82.64±1.65 | 91.03±0.20 |
| HCHA | 79.14±1.02 | 72.42±1.42 | 86.41±0.36 | 82.55±0.97 | 90.92±0.22 |
| HNHN | 76.36±1.92 | 72.64±1.57 | 86.90±0.30 | 77.19±1.49 | 86.78±0.29 |
| HyperGCN | 78.45±1.26 | 71.28±0.82 | 82.84±8.67 | 79.48±2.08 | 89.38±0.25 |
| UniGCNII | 78.81±1.05 | 73.05±2.21 | 88.25±0.40 | 83.60±1.14 | 91.69±0.19 |
| HyperND | 79.20±1.14 | 72.62±1.49 | 86.68±0.43 | 80.62±1.32 | 90.35±0.26 |
| AllDeepSets | 76.88±1.80 | 70.83±1.63 | 88.75±0.33 | 81.97±1.50 | 91.27±0.27 |
| AllSetTransformer | 78.58±1.47 | 73.08±1.20 | 88.72±0.37 | 83.63±1.47 | 91.53±0.23 |
| ED-HNN | 80.31±1.35 | 73.70±1.38 | 89.03±0.53 | 83.97±1.55 | **91.90±0.19** |
| HDS$^{ode}$ | 80.65±1.22 | 74.87±1.12 | 88.81±0.43 | 84.95±0.98 | 91.49±0.25 |
| HAMP-I | **81.18±1.30** | 75.22±1.62 | 89.02±0.38 | **85.23±1.15** | 91.66±0.17 |
| HAMP-II | 80.80±1.62 | **75.33±1.61** | **89.05±0.41** | 84.89±1.53 | 91.67±0.23 |

| Heterophilic | Congress | Senate | Walmart | House |
|---|---|---|---|---|
| HGNN | 91.26±1.15 | 48.59±4.52 | 62.00±0.24 | 61.39±2.96 |
| HCHA | 90.43±1.20 | 48.62±4.41 | 62.35±0.26 | 61.36±2.53 |
| HNHN | 53.35±1.45 | 50.93±6.33 | 47.18±0.35 | 67.80±2.59 |
| HyperGCN | 55.12±1.96 | 42.45±3.67 | 44.74±2.81 | 48.32±2.93 |
| UniGCNII | 94.81±0.81 | 49.30±4.25 | 54.45±0.37 | 67.25±2.57 |
| HyperND | 74.63±3.62 | 52.82±3.20 | 38.10±3.86 | 51.70±3.37 |
| AllDeepSets | 91.80±1.53 | 48.17±5.67 | 64.55±0.33 | 67.82±2.40 |
| AllSetTransformer | 92.16±1.05 | 51.83±5.22 | 65.46±0.25 | 69.33±2.20 |
| ED-HNN | 95.00±0.99 | 64.79±5.14 | 66.91±0.41 | 72.45±2.28 |
| HDS$^{ode}$ | 90.91±1.52 | 66.90±5.52 | 63.38±0.48 | 71.30±1.90 |
| HAMP-I | 95.09±0.79 | 69.44±6.09 | 69.90±0.38 | **72.72±1.77** |
| HAMP-II | **95.26±1.34** | **70.14±6.08** | **69.94±0.37** | 72.60±1.23 |

## 6.3 Ablation Studies

In this section, we conduct several ablation experiments on real-world datasets to assess our model design, and provide empirical validation for our theoretical findings. For more ablation experiments, see Appendix C.2.

**Impact of the Number of Layers on HAMP-I and HAMP-II.**   To assess the effectiveness of different methods in deep HNNs, we compare three representative baselines with our proposed HAMP models. Unlike Pubmed, DBLP-CA, Senate, and House, the Cora, Citeseer, and Congress datasets generally perform better in shallow networks, but worse in deep networks. Therefore, we focus on these datasets to demonstrate HAMP's advantages in deep architectures. As shown in Fig. 2, HAMP-II consistently outperforms other methods as depth increases, while competitors suffer from accuracy drops. This ability highlights the potential of HAMP to capture complex representations and maintain stability, making it a valuable framework for deep HNNs.

**Impact of the Repulsion and Allen-Cahn Forces on HAMP-I and HAMP-II.**   In addition to the theoretical analysis, we conducted ablation studies to investigate the individual and combined effects of the repulsion force $f_\beta^-$ and the Allen-Cahn force $f_d$ on both HAMP-I and HAMP-II. The results given in Tab. 2 show that incorporating the repulsive force significantly improves classification performance. In HAMP-I, enabling repulsion alone yields notable gains over the baseline, while the Allen-Cahn term alone offers moderate improvements. Notably, combining both terms consistently achieves the best accuracy across all datasets. The synergy between the repulsion and Allen-Cahn terms further boosts performance, confirming that these particle system-inspired mechanisms play complementary roles: repulsion term prevents feature over-smoothing by separating node embeddings, whereas Allen-Cahn term balances attraction and repulsion to promote class-dependent equilibrium.

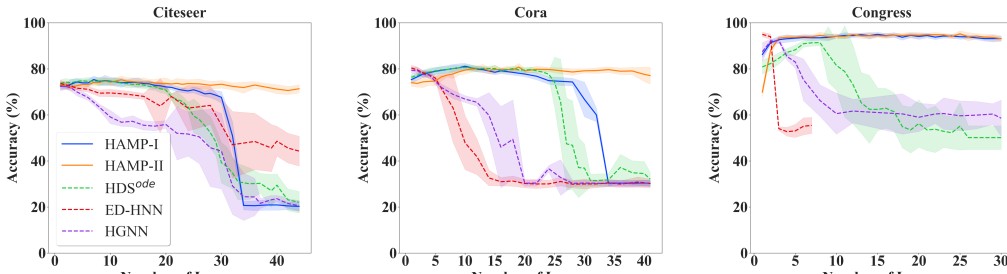

Figure 2: An empirical analysis of the depth-accuracy correlation in deep neural networks. The shaded area represents the standard deviation, helping to show the range of accuracy fluctuations.

Table 2: Node Classification on some standard hypergraph benchmarks. The accuracy (%) is reported from 10 repetitive runs. (Key: $f_\beta^-$: repulsion; $f_d$: Allen-Cahn; **Best**.)

|  | $f_\beta^-$ | $f_d$ | Cora | Citeseer | Pubmed | Congress | Senate | Walmart | House |
|---|---|---|---|---|---|---|---|---|---|
| HAMP-I | ✗ | ✗ | 75.67 | 70.59 | 87.93 | 93.47 | 60.14 | 69.86 | 69.88 |
|  | ✓ | ✗ | 75.97 | 70.60 | 88.23 | 93.65 | 61.69 | 69.73 | 69.57 |
|  | ✗ | ✓ | 80.59 | 74.67 | 88.77 | 94.67 | 65.63 | 69.73 | 71.55 |
|  | ✓ | ✓ | **81.18** | **75.22** | **89.02** | **95.09** | **69.44** | **69.90** | **72.72** |
| HAMP-II | ✗ | ✗ | 77.18 | 71.75 | 88.68 | 94.35 | 60.14 | 69.84 | 70.46 |
|  | ✓ | ✗ | 77.40 | 71.69 | 88.77 | 94.63 | 59.58 | 69.80 | 69.63 |
|  | ✗ | ✓ | 79.50 | 74.25 | 88.80 | 94.12 | 64.51 | 69.86 | 70.96 |
|  | ✓ | ✓ | **80.80** | **75.33** | **89.05** | **95.26** | **70.14** | **69.94** | **72.60** |

**Impact of Noise on HAMP-I and HAMP-II.** Fig. 3 illustrates the effect of adding stochastic component into deterministic message passing on Senate dataset. Notably, incorporating noise consistently improves accuracy for both HAMP-I and HAMP-II. In addition, the stability of HAMP-II is better than that of HAMP-I, whether or not noise is injected, indicating that the second-order particle system is more stable. Overall, these results demonstrate that incorporating stochastic component into message passing effectively improves model performance and stability by explicitly capturing data uncertainty.

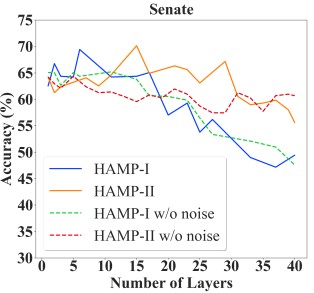

Figure 3: Significance plot for noise on Senate dataset.

## 6.4 Vertex Representation Visualization

To more intuitively validate the progressive refinement of vertex representations in our HAMP methods, we use t-SNE [39] to visualize the vertex evolution process of HAMP-I and HAMP-II on Congress dataset at different epochs. As shown in Fig. 4, we visualize the vertices based on the representations obtained at epoch $1, \frac{1}{8}E, \frac{1}{4}E, \frac{1}{2}E$, and $E$, where $E$ is the total number of epochs. From Fig. 4, we have the following three observations:

- When epoch = 1 (subgraphs (a) and (f)), the node feature representations exhibit a chaotic distribution, and it is difficult to distinguish the number of categories. As training progresses, the clustering entropy shows a monotonically decreasing trend.

- Subgraphs (e) and (j) show the visualization results at convergence for HAMP-I and HAMP-II, respectively. The final category boundaries are clearer in HAMP-II than in HAMP-I, reflecting the geometry refinement enabled by HAMP-II's deeper message passing.

- Comparing subgraphs (b) and (g), HAMP-I is still in the early stages of categorization, while HAMP-II shows a significantly improved clustering effect. Notably, HAMP-II has successfully distinguished six distinct categories, confirming that HAMP-II achieves faster cluster than HAMP-I due to the second-order mechanism.

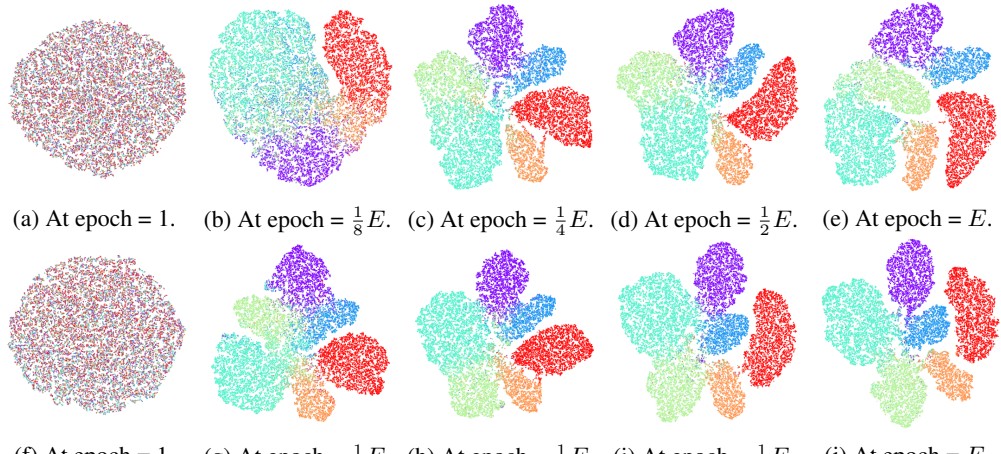

(a) At epoch = 1.  (b) At epoch = $\frac{1}{8}E$.  (c) At epoch = $\frac{1}{4}E$.  (d) At epoch = $\frac{1}{2}E$.  (e) At epoch = $E$.

(f) At epoch = 1.  (g) At epoch = $\frac{1}{8}E$.  (h) At epoch = $\frac{1}{4}E$.  (i) At epoch = $\frac{1}{2}E$.  (j) At epoch = $E$.

Figure 4: The t-SNE visualization of vertex representation evolution of HAMP-I (the first row) and HAMP-II (the second row) on Congress dataset. The colors represent the class labels.

## 7 Related Work

**Hypergraph Neural Networks.** Hypergraph learning was first introduced in [51] as a propagation process on hypergraph. Since then, hypergraph learning [21, 28] has developed extensively. As an extension of GNNs, Feng et al. [20] proposed HGNN that effectively captures high-order interactions by leveraging the vertex-edge-vertex propagation pattern. Further, HGNN$^+$ [22] introduced hyperedge groups and adaptive hyperedge group fusion strategy as a general framework for modeling high-order multi-modal/multi-type data correlations. Following the spectral theory of hypergraph [47], SheafHyperGNN [16] introduced the sheaf theory to model data relationships in hypergraph more finely. Inspired by Transformers [40], several HNNs [2, 11, 12] enhanced the feature extraction capability for hypergraph through attention mechanism and centrality for positional encoding. Different from using vertex-vertex propagation pattern, several works [11, 26, 43, 48] have considered employing multi-phase message passing. Among these, UniGNN [26] presented a unified framework that that facilitates the processing of all hypergraph data through GNNs. In contrast, ED-HNN [43] and CoNHD [49] were developed from the perspective of optimization of hyperedge and node potential. Additionally, HDS$^{ode}$ [48] adopted control-diffusion ODEs to model the hypergraph dynamic system. By contrast, our method is based on the particle system theory, employing the first-order and second-order systems to understand and design HNNs.

**Collective Dynamics.** Watts and Strogatz [45] first mathematically defined a small-world network and explained the reasons behind the collective dynamics. Many researchers are interested in how active/passive media made of many interacting agents form complex patterns in mathematical biology and technology. Battiston et al. [4] showed that higher-order interactions play a crucial role in understanding these complex patterns. These complex patterns can be seen in animal groups, cell clusters, granular media, and self-organizing particles, as shown in [7, 25, 31] and other references. In many of these models, the agents move into groups based on the attractive-repulsive forces [8, 17, 6]. For example, Fang et al. [19] have studied the Cucker-Smale model [13] with Rayleigh friction and attractive-repulsive coupling, and [27] showed a similar collective phenomenon with stochastic dynamics. Furthermore, the ACMP [44] was based on Allen-Cahn particle system and incorporated the repulsive force, providing inspiration for our work in hypergraph learning.

## 8 Conclusion

In this paper, we introduce a novel hypergraph message passing framework inspired by particle system theory. We derive both first-order and second-order system equations, yielding two distinct models for modeling hypergraph message passing dynamics that capture full hyperedge interactions while mitigating over-smoothing and heterophily. The proposed models further integrate a stochastic term to model uncertainty in these interactions and can alleviate over-smoothing in deep layers. By casting

HNNs in a physically interpretable paradigm, our model balances high-order interaction modeling with feature-diversity preservation, offering both theoretical insights and practical advances for complex system analysis. In future work, we plan to extend the proposed methods to protein-structure and sequence design for the discovery of novel antibodies and enzymes.

## Acknowledgments and Disclosure of Funding

This work is supported by the National Science Foundation for International Senior Scientists Grant (No. 12350710181), and the Shanghai Municipal Science and Technology Key Project (No. 22JC1402300). We thank the anonymous reviewers for their insightful comments.

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

# A  Algorithms

**Complexity Analysis.**    Here, we analyze the computational complexity of one layer in HAMP-I and HAMP-II. Analytically, the time complexity is $\mathcal{O}\left(|\mathcal{V}||\mathcal{E}|c^2 + |\mathcal{V}|c\right)$, where $|\mathcal{V}|$, $|\mathcal{E}|$ and $c$ are the number of nodes, number of hyperedges and number of hidden dimension, respectively. However, the incidence matrix $\mathbf{H}$ is a sparse matrix, so the time complexity is $\mathcal{O}\left((tr(\mathbf{D}_v) + tr(\mathbf{D}_e))c^2 + |\mathcal{V}|c\right)$, where $tr(\mathbf{D}_v)$ is the sum of the degrees of all nodes and $tr(\mathbf{D}_e)$ is the sum of the number of nodes contained in all hyperedges. The detailed process of HAMP-I and HAMP-II are shown in Algorithm 1 and Algorithm 2.

---

**Algorithm 1** The HAMP-I Algorithm for Hypergraph Node Classification.

---
1:  **Input**: the incidence matrix $\mathbf{H}$, the node feature $\mathbf{X}$, and the node labels $\mathbf{Y}$ .
2:  **Output**: the model prediction accuracy .
3:  **Initialization**: the time $T$, the step size $\tau$ and all parameters of model .
4:  **while** not converged **do**
5:      Node feature mapping $\mathbf{X} = \text{Linear}_{\text{map}}(\mathbf{X})$ ;
6:      Set the initial time $t = 0$, the initial node representation $\mathbf{X}(0) = \mathbf{X}$ ;
7:      **while** $t \leq T$ **do**
8:          Message passing from $\mathcal{V}$ to $\mathcal{E}$: $\mathbf{X}_{\mathcal{V} \rightarrow \mathcal{E}}(t) = \Phi_1(\mathbf{X}(t), \mathbf{H})$ ;
9:          Message passing from $\mathcal{E}$ to $\mathcal{V}$: $\mathbf{X}_{\mathcal{E} \rightarrow \mathcal{V}}(t) = \Psi\left(\mathbf{X}(t), \Phi_2\left(\mathbf{X}_{\mathcal{V} \rightarrow \mathcal{E}}(t), \mathbf{H}\right)\right)$ ;
10:          Compute particle dynamics as

$$\mathbf{X}(t+\tau) = \mathbf{X}(t) + \tau\sigma\left(\underbrace{\mathbf{X}_{\mathcal{E}\rightarrow\mathcal{V}}(t) - \omega\mathbf{X}(t)}_{\text{Interaction Force}} + \underbrace{\delta f_d(\mathbf{X}(t))}_{\text{Allen-Cahn Force}} + \underbrace{\epsilon\mathbf{B}(t)}_{\text{Noise}} + \beta\mathbf{X}(0)\right) ;$$

11:          Updata $t = t + \tau$ ;
12:      **end while**
13:      Input the node representation into the classifier $\mathbf{X}^{out} = \text{MLP}(\mathbf{X}(T))$;
14:      Compute the model prediction labels $\widehat{\mathbf{Y}} = \text{Softmax}(\mathbf{X}^{out})$ and compute the loss function ;
15:      Update all parameter by back propagation using the Adam optimizer ;
16: **end while**

---

---

**Algorithm 2** The HAMP-II Algorithm for Hypergraph Node Classification.

---
1:  **Input**: the incidence matrix $\mathbf{H}$, the node feature $\mathbf{X}$, and the node labels $\mathbf{Y}$ .
2:  **Output**: the model prediction accuracy .
3:  **Initialization**: the time $T$, the step size $\tau$ and all parameters of model .
4:  **while** not converged **do**
5:      Node feature mapping $\mathbf{X} = \text{Linear}_{\text{map}}(\mathbf{X})$ ;
6:      Set the initial time $t = 0$, the initial node representation $\mathbf{X}(0) = \mathbf{X}$ ;
7:      Set the initial velocity $\mathbf{V}(0) = \text{Linear}(\mathbf{X}(0)) - \mathbf{X}(0)$ ;
8:      **while** $t \leq T$ **do**
9:          Message passing from $\mathcal{V}$ to $\mathcal{E}$: $\mathbf{V}_{\mathcal{V} \rightarrow \mathcal{E}}(t) = \Phi_1(\mathbf{V}(t), \mathbf{H})$ ;
10:          Message passing from $\mathcal{E}$ to $\mathcal{V}$: $\mathbf{V}_{\mathcal{E} \rightarrow \mathcal{V}}(t) = \Psi\left(\mathbf{V}(t), \Phi_2\left(\mathbf{V}_{\mathcal{V} \rightarrow \mathcal{E}}(t), \mathbf{H}\right)\right)$ ;
11:          Compute the velocity of particle dynamics system as

$$\mathbf{V}(t+\tau) = \mathbf{V}(t) + \tau\sigma\left(\underbrace{\mathbf{V}_{\mathcal{E}\rightarrow\mathcal{V}}(t) - \omega\mathbf{V}(t)}_{\text{Interaction Force}} + \underbrace{\delta f_d(\mathbf{V}(t))}_{\text{Allen-Cahn Force}} + \underbrace{\epsilon\mathbf{B}(t)}_{\text{Noise}} + \beta\mathbf{V}(0)\right) ;$$

12:          Compute the representation by $\mathbf{X}(t+\tau) = \mathbf{X}(t) + \tau\mathbf{V}(t+\tau)$ ;
13:          Updata $t = t + \tau$ ;
14:      **end while**
15:      Input the node representation into the classifier $\mathbf{X}^{out} = \text{MLP}(\mathbf{X}(T))$;
16:      Compute the model prediction labels $\widehat{\mathbf{Y}} = \text{Softmax}(\mathbf{X}^{out})$ and compute the loss function ;
17:      Update all parameter by back propagation using the Adam optimizer ;
18: **end while**

---

**Limitations Discussion.** Our particle dynamics-based hypergraph message passing framework assumes a static hypergraph topology. While this assumption is valid for social and biological hypergraphs with slowly evolving interactions, it may not hold in highly dynamic scenarios like financial transaction networks, where hyperedge topologies change abruptly. Consequently, effectively modeling temporal hypergraphs with evolving structures remains an open challenge.

# B  Theoretical Results and Proof

Technically, we suppose there exists $\{f_\beta^e\}$ such that $\mathcal{I} = \{1, \cdots, N\}$ can be divided into two disjoint groups with $N_1, N_2$ particles respectively: $f_\beta(h_{i,j}^e) \geq 0$, for $\{i, j\} \in \mathcal{I}_1$ or $\mathcal{I}_2$ and $f_\beta(h_{i,j}^e) \leq 0$, otherwise. We designate

$$\{x_i^{(1)}\} := \{\mathbf{x}_i | i \in \mathcal{I}_1\}, \quad \{x_j^{(2)}\} := \{\mathbf{x}_j | j \in \mathcal{I}_2\}. \tag{11}$$

The model is channel-wise, hence we use $x$ instead of $\mathbf{x}$ in the proof.

First, let's define the relevant notation,

The mean value:

$$\bar{x} := \frac{1}{N} \sum_{i=1}^{N} x_i. \tag{12}$$

The deviation values:

$$\hat{x}_i := x_i - \bar{x}. \tag{13}$$

The variance of values within each group:

$$\mathrm{var}(x^{(1)}) = \frac{1}{N_1} \sum (\hat{x}_i^{(1)})^2, \quad \mathrm{var}(x^{(2)}) = \frac{1}{N_2} \sum (\hat{x}_i^{(2)})^2. \tag{14}$$

The second moments:

$$M_2(x^{(1)}) := \sum_{i=1}^{N_1} (x_i^{(1)})^2, \quad M_2(x^{(2)}) := \sum_{i=1}^{N_2} (x_i^{(2)})^2. \tag{15}$$

And others:

$$\widehat{M_2} := M_2(\hat{x}^{(1)}) + M_2(\hat{x}^{(2)}) = \mathrm{var}(x^{(1)}) + \mathrm{var}(x^{(2)}). \tag{16}$$

$$\psi_i^{e,\pm} := \sum_{j \in e} h_{i,j}^{e,\pm}, \quad \psi_i^{\pm} := \sum_{e \in \mathcal{E}} \sum_{j \in e} h_{i,j}^{e,\pm}. \tag{17}$$

$$k := \max_i \{|\mathcal{E}(i)|\}. \tag{18}$$

$$D_e^- := \max_k \{\psi_k^{e,-}\}, \ D^- := \max_e \{D_e^-\}. \tag{19}$$

$$D_2^- := \max_e \{\|\psi^{e,-}\|_2\}. \tag{20}$$

Technically, we set $N_1 = N_2 := N_0$. This assumption is that $N_1$ is comparable to $N_2$, i.e., there exists a positive constant $\kappa$ satisfying $\frac{1}{\kappa} N_1 \leq N_2 \leq \kappa N_1$.

We can rewrite Eq. 4 as

$$\begin{cases} \dot{u}_i^{(1)} = \dfrac{1}{N_1} \displaystyle\sum_{e \in \mathcal{E}(i)} \sum_{i'=1}^{N_1} h_{i,i'}^{e,+} (x_{i'}^{(1)} - x_i^{(1)}) - \dfrac{1}{N_2} \sum_{e \in \mathcal{E}(i)} \sum_{j=1}^{N_2} h_{i,j}^{e,-} (x_j^{(2)} - x_i^{(1)}) + \delta x_i^{(1)} (1 - (x_i^{(1)})^2) \\[4mm] \dot{v}_j^{(2)} = \dfrac{1}{N_2} \displaystyle\sum_{e \in \mathcal{E}(j)} \sum_{j'=1}^{N_2} h_{j,j'}^{e,+} (x_{j'}^{(2)} - x_j^{(2)}) - \dfrac{1}{N_1} \sum_{e \in \mathcal{E}(j)} \sum_{i=1}^{N_1} h_{i,j}^{e,-} (x_i^{(1)} - x_j^{(2)}) + \delta x_j^{(2)} (1 - (x_j^{(2)})^2). \end{cases} \tag{21}$$

Set the matrix $A^e$ as

$$A^e_{i,j} = \begin{cases} h^e_{i,i} \\ -h^e_{i,j}, \end{cases} \tag{22}$$

and and designate $C^A := \min_{e \in \mathcal{E}} \{F(A^e)\}$, where $F(A^e)$ is the *Fiedler number* of $A^e$.

**Lemma B.1** ($L_2$ estimate for $M_2$)**.** *There exists a positive constant $M_2^\infty$ such that*

$$\sup_{0 \leq t < \infty} M_2(t) \leq M_2^\infty \leq \infty. \tag{23}$$

*Proof.* Note that $h^{e,\pm}_{i,j} = h^{e,\pm}_{j,i}$, then

$$\begin{aligned}
\frac{\mathrm{d}}{\mathrm{d}t} M_2(x^{(1)}) &= \frac{2}{N_1} \sum_{i=1}^{N_1} x_i^{(1)} \dot{x}_i^{(1)} \\
&= \frac{2}{N_1} \sum_{i=1}^{N_1} \sum_{e \in \mathcal{E}(i)} \sum_{i' \in e} h^{e,+}_{i,i'} (x_{i'}^{(1)} - x_i^{(1)}) x_i^{(1)} - \frac{2}{N_1} \sum_{i=1}^{N_1} \sum_{e \in \mathcal{E}(i)} \sum_{j \in e} h^{e,-}_{i,j} (x_j^{(2)} - x_i^{(1)}) x_i^{(1)} \\
&\quad + \frac{2\delta}{N_1} \sum_{i=1}^{N_1} (x_i^{(1)})^2 (1 - (x_i^{(1)})^2) \\
&= \frac{2}{N_1} \sum_{e \in \mathcal{E}} \sum_{i,i' \in e} h^{e,+}_{i,i'} (x_{i'}^{(1)} - x_i^{(1)}) x_i^{(1)} - \frac{2}{N_1} \sum_{e \in \mathcal{E}} \sum_{i,j \in e} h^{e,-}_{i,j} (x_j^{(2)} - x_i^{(1)}) x_i^{(1)} \\
&\quad + \frac{2\delta}{N_1} \sum_{i=1}^{N_1} (x_i^{(1)})^2 (1 - (x_i^{(1)})^2) \\
&= -\frac{1}{N_1} \sum_{e \in \mathcal{E}} \sum_{i,i' \in e} h^{e,+}_{i,i'} (x_{i'}^{(1)} - x_i^{(1)})^2 - \frac{2}{N_1} \sum_{e \in \mathcal{E}} \sum_{i,j \in e} h^{e,-}_{i,j} (x_j^{(2)} - x_i^{(1)}) x_i^{(1)} \\
&\quad + \frac{2\delta}{N_1} \sum_{i=1}^{N_1} (x_i^{(1)})^2 (1 - (x_i^{(1)})^2).
\end{aligned} \tag{24}$$

Similarly,

$$\begin{aligned}
\frac{\mathrm{d}}{\mathrm{d}t} M_2(x^{(2)}) &= \frac{2}{N_2} \sum_{j=1}^{N_2} x_j^{(2)} \dot{x}_j^{(2)} \\
&= -\frac{1}{N_2} \sum_{e \in \mathcal{E}} \sum_{j,j' \in e} h^{e,+}_{j,j'} (x_{j'}^{(2)} - x_j^{(2)})^2 - \frac{2}{N_2} \sum_{e \in \mathcal{E}} \sum_{j,i \in e} h^{e,-}_{j,i} (x_i^{(1)} - x_j^{(2)}) x_j^{(2)} \\
&\quad + \frac{2\delta}{N_2} \sum_{j=1}^{N_2} (x_j^{(2)})^2 (1 - (x_j^{(2)})^2).
\end{aligned} \tag{25}$$

Define the total second moment $M_2 := M_2(x^{(1)}) + M_2(x^{(2)})$. Its time derivative is:

$$\frac{\mathrm{d}}{\mathrm{d}t} M_2 = \frac{\mathrm{d}}{\mathrm{d}t} M_2(x^{(1)}) + \frac{\mathrm{d}}{\mathrm{d}t} M_2(x^{(2)}). \tag{26}$$

By discarding the non-positive squared terms (first two sums), we obtain the inequality:

$$\begin{aligned}
\frac{\mathrm{d}}{\mathrm{d}t} M_2 \leq &-\frac{2}{N_1} \sum_{e \in \mathcal{E}} \sum_{i,j \in e} h^{e,-}_{i,j} (x_j^{(2)} - x_i^{(1)}) x_i^{(1)} - \frac{2}{N_2} \sum_{e \in \mathcal{E}} \sum_{j,i \in e} h^{e,-}_{j,i} (x_i^{(1)} - x_j^{(2)}) x_j^{(2)} \\
&+ \frac{2\delta}{N_1} \sum_{i=1}^{N_1} (x_i^{(1)})^2 (1 - (x_i^{(1)})^2) + \frac{2\delta}{N_2} \sum_{i=1}^{N_2} (x_i^{(2)})^2 (1 - (x_i^{(2)})^2).
\end{aligned} \tag{27}$$

By the Cauchy-Schwarz inequality,

$$(M_2^{(1)})^2 = \left(\sum_{i=1}^{N_1}(x_i^{(1)})^2\right)^2 \leq N_1 \sum_{i=1}^{N_1}(x_i^{(1)})^4,$$

$$(M_2^{(2)})^2 = \left(\sum_{i=1}^{N_2}(x_i^{(2)})^2\right)^2 \leq N_2 \sum_{i=1}^{N_2}(x_i^{(2)})^4,$$

$$(x_i^{(1)} - x_j^{(2)})^2 \leq 2((x_i^{(1)})^2 + (x_j^{(2)})^2).$$

Then we have

$$\frac{2\delta}{N_1}\sum_{i=1}^{N_1}(x_i^{(1)})^2\left(1-(x_i^{(1)})^2\right) + \frac{2\delta}{N_2}\sum_{i=1}^{N_2}(x_i^{(2)})^2\left(1-(x_i^{(2)})^2\right)$$

$$=\frac{2\delta}{N_1}\sum_{i=1}^{N_1}(x_i^{(1)})^2 - \frac{2\delta}{N_1}\sum_{i=1}^{N_1}(x_i^{(1)})^4 + \frac{2\delta}{N_2}\sum_{i=1}^{N_2}(x_i^{(2)})^2 - \frac{2\delta}{N_2}\sum_{i=1}^{N_2}(x_i^{(2)})^4$$

$$\leq\frac{2\delta}{N_1}\sum_{i=1}^{N_1}(x_i^{(1)})^2 - \frac{2\delta}{(N_1)^2}\left(\sum_{i=1}^{N_1}(x_i^{(1)})^2\right)^2 + \frac{2\delta}{N_2}\sum_{i=1}^{N_2}(x_i^{(2)})^2 - \frac{2\delta}{(N_2)^2}\left(\sum_{i=1}^{N_2}(x_i^{(2)})^2\right)^2 \tag{28}$$

$$=2\delta\left(\frac{M_2^{(1)}}{N_1} + \frac{M_2^{(2)}}{N_2}\right) - 2\delta\left(\left(\frac{M_2^{(1)}}{N_1}\right)^2 + \left(\frac{M_2^{(2)}}{N_2}\right)^2\right)$$

$$\leq 2\delta\left(\frac{M_2^{(1)}}{N_1} + \frac{M_2^{(2)}}{N_2}\right) - \delta\left(\frac{M_2^{(1)}}{N_1} + \frac{M_2^{(2)}}{N_2}\right)^2$$

$$\leq\frac{2\delta}{N'}M_2 - \frac{\delta}{N''}(M_2)^2,$$

where $N' = \min\{N_1, N_2\}$ and $N'' = \max\{N_1, N_2\}$.

So, we have

$$\frac{\mathrm{d}}{\mathrm{d}t}M_2 \leq \frac{D^-}{N_1}\sum_{e\in\mathcal{E}}\sum_{i,j\in e}\left((x_j^{(2)} - x_i^{(1)})^2 + (x_i^{(1)})^2\right) + \frac{D^-}{N_2}\sum_{e\in\mathcal{E}}\sum_{j,i\in e}\left((x_i^{(1)} - x_j^{(2)})^2 + (x_j^{(2)})^2\right)$$

$$+ \frac{2\delta}{N_1}\sum_{i=1}^{N_1}(x_i^{(1)})^2\left(1-(x_i^{(1)})^2\right) + \frac{2\delta}{N_2}\sum_{i=1}^{N_2}(x_i^{(2)})^2\left(1-(x_i^{(2)})^2\right)$$

$$\leq\frac{D^-}{N_1}\sum_{e\in\mathcal{E}}\sum_{i,j\in e}\left(3(x_i^{(1)})^2 + 2(x_j^{(2)})^2\right) + \frac{D^-}{N_2}\sum_{e\in\mathcal{E}}\sum_{i,j\in e}\left(2(x_i^{(1)})^2 + 3(x_j^{(2)})^2\right) + \frac{2\delta}{N'}M_2 - \frac{\delta}{N''}M_2^2. \tag{29}$$

These relations yield a Riccati-type differential inequality:

$$\frac{\mathrm{d}}{\mathrm{d}t}M_2 \leq 5D^-\max\{\frac{1}{N_1}, \frac{1}{N_2}\}\sum_{e\in\mathcal{E}}\sum_{i,j\in e}\left((x_i^{(1)})^2 + (x_j^{(2)})^2\right) + 2\delta M_2 - \delta(M_2)^2$$

$$\leq\frac{5D^-k}{N'}M_2 + \frac{2\delta}{N'}M_2 - \frac{\delta}{N''}(M_2)^2 \tag{30}$$

$$\leq\frac{5D^-k + 2\delta}{N'}M_2 - \frac{\delta}{N''}(M_2)^2.$$

Let $y$ be a solution of the following ODE:

$$y' = ay - by^2. \tag{31}$$

Then, by phase line analysis, the solution $y(t)$ to Eq. 31 satisfies

$$M_2(t) \leq y(t) \leq \max\left\{\frac{a}{b}, M_2(0)\right\} = \max\left\{\frac{(5D^-k + 2\delta)N''}{\delta N'}, M_2(0)\right\} =: M_2^\infty. \tag{32}$$

which yields the desired estimate. ∎

**Proposition B.2.** *For Eq. 4, the distance of the centers of the two clusters is finite.*

*Proof of Propostion B.2.* By Lemma B.1

$$\|\bar{x}^{(1)} - \bar{x}^{(2)}\| = \left\| \frac{1}{N_1} \sum_{i=1}^{N_1} x_i^{(1)} - \frac{1}{N_2} \sum_{j=1}^{N_2} x_j^{(2)} \right\|$$

$$\leq \sqrt{2} \sqrt{\frac{1}{N_1} \sum_{i=1}^{N_1} (x_i^{(1)})^2 + \frac{1}{N_2} \sum_{j=1}^{N_2} (x_j^{(2)})^2} \tag{33}$$

$$\leq \sqrt{2} \sqrt{M_2^\infty},$$

where the first inequality used the Cauchy-Schwarz inequality. ∎

**Lemma B.3.** *Let $u, v$ be the solution to Eq. 21. Then $\|\bar{x}^{(1)} - \bar{x}^{(2)}\|^2$ satisfies*

$$\frac{1}{2} \frac{\mathrm{d}}{\mathrm{d}t} \|\bar{x}^{(1)} - \bar{x}^{(2)}\|^2 \geq \left( \frac{2c_m}{N_0} - c_1 \right) \|\bar{x}^{(1)} - \bar{x}^{(2)}\|^2 - \frac{4(D_2^-)^2}{c_1 N_0} k \widehat{M}_2. \tag{34}$$

*Proof.* The time evolution of $\bar{x}^{(1)}$ is given by

$$\dot{\bar{x}}^{(1)} = \frac{1}{N_1} \sum_{i=1}^{N_1} \sum_{i' \in \mathcal{E}(i)} h_{i,i'}^{e,+} (x_{i'}^{(1)} - x_i^{(1)}) - \frac{1}{N_1} \sum_{i=1}^{N_1} \sum_{j \in \mathcal{E}(i)} h_{j,i}^{e,-} (x_j^{(2)} - x_i^{(1)})$$

$$= -\frac{1}{N_1} \sum_{e \in \mathcal{E}} \sum_{i,j \in e} h_{i,j}^{e,-} (x_j^{(2)} - x_i^{(1)})$$

$$= -\frac{1}{N_1} \sum_{e \in \mathcal{E}} \psi_j^{e,-} x_j^{(2)} + \frac{1}{N_1} \sum_{e \in \mathcal{E}} \psi_i^{e,-} x_i^{(1)} \tag{35}$$

$$= \sum_{e \in \mathcal{E}} \left( -\frac{1}{N_1} \psi_j^{e,-} x_j^{(2)} + \frac{1}{N_1} \psi_i^{e,-} x_i^{(1)} \right),$$

where the first equality uses the relation $\sum_{i=1}^{N_1} \hat{x}_i^{(1)} = 0$.

Then we have

$$\frac{1}{2} \frac{\mathrm{d}}{\mathrm{d}t} \|\bar{x}^{(1)} - \bar{x}^{(2)}\|^2 \tag{36}$$

$$= (\bar{x}^{(1)} - \bar{x}^{(2)})(\dot{\bar{x}}^{(1)} - \dot{\bar{x}}^{(2)}) \tag{37}$$

$$= (\bar{x}^{(1)} - \bar{x}^{(2)}) \left[ \sum_{e \in \mathcal{E}} \left( -\frac{1}{N_1} \psi_j^{e,-} x_j^{(2)} + \frac{1}{N_1} \psi_i^{e,-} x_i^{(1)} \right) - \sum_{e \in \mathcal{E}} \left( -\frac{1}{N_2} \psi_i^{e,-} x_i^{(1)} + \frac{1}{N_2} \psi_j^{e,-} x_j^{(2)} \right) \right] \tag{38}$$

$$= (\bar{x}^{(1)} - \bar{x}^{(2)}) \sum_{e \in \mathcal{E}} \left( -\frac{1}{N_1} \psi_j^{e,-} x_j^{(2)} - \frac{1}{N_2} \psi_j^{e,-} x_j^{(2)} + \frac{1}{N_1} \psi_i^{e,-} x_i^{(1)} + \frac{1}{N_2} \psi_i^{e,-} x_i^{(1)} \right) \tag{39}$$

$$= (\bar{x}^{(1)} - \bar{x}^{(2)}) \sum_{e \in \mathcal{E}} \Big( -\frac{1}{N_1} \psi_j^{e,-} (\bar{x}^{(2)} + \hat{x}_j^{(2)}) - \frac{1}{N_2} \psi_j^{e,-} (\bar{x}^{(2)} + \hat{x}_j^{(2)}) + \frac{1}{N_1} \psi_i^{e,-} (\bar{x}^{(1)} + \hat{x}_i^{(1)}) \tag{40}$$

$$+ \frac{1}{N_2} \psi_i^{e,-} (\bar{x}^{(1)} + \hat{x}_j^{(1)}) \Big) \tag{41}$$

$$= (\bar{x}^{(1)} - \bar{x}^{(2)}) \left\{ \left[ -\left( \frac{1}{N_1} + \frac{1}{N_2} \right) \sum_{e \in \mathcal{E}} \sum_{j \in e} \psi_j^{e,-} \right] \bar{x}^{(2)} + \left[ \left( \frac{1}{N_1} + \frac{1}{N_2} \right) \sum_{e \in \mathcal{E}} \sum_{i \in e} \psi_i^{e,-} \right] \bar{x}^{(1)} \right. \tag{42}$$

$$\left[-\left(\frac{1}{N_1}+\frac{1}{N_2}\right)\sum_{e\in\mathcal{E}}\sum_{j\in e}\psi_j^{e,-}\right]\hat{x}_j^{(2)}+\left[\left(\frac{1}{N_1}+\frac{1}{N_2}\right)\sum_{e\in\mathcal{E}}\sum_{i\in e}\psi_i^{e,-}\right]\hat{x}_i^{(1)}\right\}\tag{43}$$

$$=\frac{2}{N_0}(\bar{x}^{(1)}-\bar{x}^{(2)})\left[\sum_{e\in\mathcal{E}}\sum_{i\in e}\psi_i^{e,-}\bar{x}^{(1)}-\sum_{e\in\mathcal{E}}\sum_{j\in e}\psi_j^{e,-}\bar{x}^{(2)}\right]\tag{44}$$

$$+\frac{2}{N_0}(\bar{x}^{(1)}-\bar{x}^{(2)})\left[\sum_{e\in\mathcal{E}}\sum_{i\in e}\psi_i^{e,-}\hat{x}_i^{(1)}-\sum_{e\in\mathcal{E}}\sum_{j\in e}\psi_j^{e,-}\hat{x}_j^{(2)}\right].\tag{45}$$

We denote

$$\mathrm{Pes}(\hat{x}^{(1)},\hat{x}^{(2)}):=\sum_{e\in\mathcal{E}}\sum_{i\in e}\psi_i^{e,-}\hat{x}_i^{(1)}-\sum_{e\in\mathcal{E}}\sum_{j\in e}\psi_j^{e,-}\hat{x}_j^{(2)}.$$

and

$$\mathrm{Pes}(\bar{x}^{(1)},\bar{x}^{(2)}):=\sum_{e\in\mathcal{E}}\sum_{i\in e}\psi_i^{e,-}\bar{x}^{(1)}-\sum_{e\in\mathcal{E}}\sum_{j\in e}\psi_j^{e,-}\bar{x}^{(2)}.$$

Assume there exist constants $c_m, c_v$, such that

$$\mathrm{Pes}(\bar{x}^{(1)},\bar{x}^{(2)})\geq c_m(\bar{x}^{(1)}-\bar{x}^{(2)}).\tag{46}$$

Then, by Cauchy's inequality, for any $c_1$, we have

$$\frac{1}{2}\frac{\mathrm{d}}{\mathrm{d}t}\|\bar{x}^{(1)}-\bar{x}^{(2)}\|^2$$

$$=\frac{2}{N_0}\mathrm{Pes}(\bar{x}^{(1)},\bar{x}^{(2)})(\bar{x}^{(1)}-\bar{x}^{(2)})+\frac{2}{N_0}\mathrm{Pes}(\hat{x}^{(1)},\hat{x}^{(2)})(\bar{x}^{(1)}-\bar{x}^{(2)})$$

$$\geq\left(\frac{2c_m}{N_0}-c_1\right)\|\bar{x}^{(1)}-\bar{x}^{(2)}\|^2+c_1\|\bar{x}^{(1)}-\bar{x}^{(2)}\|^2+\frac{2}{N_0}\mathrm{Pes}(\hat{x}^{(1)},\hat{x}^{(2)})(\bar{x}^{(1)}-\bar{x}^{(2)})$$

$$\geq\left(\frac{2c_m}{N_0}-c_1\right)\|\bar{x}^{(1)}-\bar{x}^{(2)}\|^2-\frac{1}{c_1}\frac{2}{N_0}\left(\mathrm{Pes}(\hat{x}^{(1)},\hat{x}^{(2)})\right)^2$$

$$\geq\left(\frac{2c_m}{N_0}-c_1\right)\|\bar{x}^{(1)}-\bar{x}^{(2)}\|^2-\frac{4}{c_1N_0}\sum_{e\in\mathcal{E}}\left[\left(\sum_{i\in e}\psi_i^{e,-}\hat{x}_i^{(1)}\right)^2+\left(\sum_{j\in e}\psi_j^{e,-}\hat{x}_j^{(2)}\right)^2\right]\tag{47}$$

$$\geq\left(\frac{2c_m}{N_0}-c_1\right)\|\bar{x}^{(1)}-\bar{x}^{(2)}\|^2-\frac{4}{c_1N_0}\sum_{e\in\mathcal{E}}\left[\|\psi^{e,-}\|^2\sum_{i\in e}(\hat{x}_i^{(1)})^2+\|\psi^{e,-}\|^2\sum_{j\in e}(\hat{x}_j^{(2)})^2\right]$$

$$\geq\left(\frac{2c_m}{N_0}-c_1\right)\|\bar{x}^{(1)}-\bar{x}^{(2)}\|^2-\frac{4(D_2^-)^2}{c_1N_0}\sum_{e\in\mathcal{E}}\left[\sum_{i\in e}(\hat{x}_i^{(1)})^2+\sum_{j\in e}(\hat{x}_j^{(2)})^2\right]$$

$$\geq\left(\frac{2c_m}{N_0}-c_1\right)\|\bar{x}^{(1)}-\bar{x}^{(2)}\|^2-\frac{4(D_2^-)^2}{c_1N_0}k[\widehat{M}_2].$$

∎

**Lemma B.4.** *Let $u, v$ be the solution to Eq. 21. Then $\widehat{M}_2$ satisfies*

$$\frac{1}{2}\frac{\mathrm{d}}{\mathrm{d}t}\widehat{M}_2\leq C_2\widehat{M}_2+2c_2\|\bar{x}^{(1)}-\bar{x}^{(2)}\|^2,\tag{48}$$

*where*

$$C_2:=-k\left(C^A-\frac{D^-}{2}+\frac{(D^-)^2}{4c_2}-\frac{\delta}{k}\right),\tag{49}$$

*and $c_2$ is an arbitrary positive constant.*

*Proof.* Subtracting Eq. 35 from Eq. 21 gives $\dot{\hat{x}}_i^{(1)}$. Then we have

$$\frac{1}{2}\frac{\mathrm{d}}{\mathrm{d}t}\left(\frac{1}{N_1}\sum_{i=1}^{N_1}\|\hat{x}_i^{(1)}\|^2\right)$$

$$=\frac{1}{N_1}\sum_{i=1}^{N_1}\hat{x}_i^{(1)}\dot{\hat{x}}_i^{(1)}$$

$$=\frac{1}{N_1}\sum_{i=1}^{N_1}\hat{x}_i^{(1)}\sum_{e\in\mathcal{E}(i)}\left[\sum_{i'\in e}h_{i,i'}^{e,+}(x_{i'}^{(1)}-x_i^{(1)})-\sum_{j\in e}h_{i,j}^{e,-}(x_j^{(2)}-x_i^{(1)})\right]$$

$$+\frac{\delta}{N_1}\sum_{i=1}^{N_1}\hat{x}_i^{(1)}x_i^{(1)}(1-(x_i^{(1)})^2)$$

$$=\frac{1}{N_1}\sum_{i=1}^{N_1}\hat{x}_i^{(1)}\sum_{e\in\mathcal{E}(i)}\left[\sum_{i'\in e}h_{i,i'}^{e,+}(\hat{x}_{i'}^{(1)}-\hat{x}_i^{(1)})-\sum_{j\in e}h_{i,j}^{e,-}(x_j^{(2)}-x_i^{(1)})\right] \qquad (50)$$

$$+\frac{\delta}{N_1}\sum_{i=1}^{N_1}\hat{x}_i^{(1)}x_i^{(1)}(1-(x_i^{(1)})^2)$$

$$=\frac{1}{N_1}\sum_{i=1}^{N_1}\hat{x}_i^{(1)}\sum_{e\in\mathcal{E}(i)}\sum_{i'\in e}h_{i,i'}^{e,+}(\hat{x}_{i'}^{(1)}-\hat{x}_i^{(1)})-\frac{1}{N_1}\sum_{i=1}^{N_1}\hat{x}_i^{(1)}\sum_{e\in\mathcal{E}(i)}\sum_{j\in e}h_{i,j}^{e,-}(\hat{x}_j^{(2)}-\hat{x}_i^{(1)})$$

$$-\frac{1}{N_1}\sum_{i=1}^{N_1}\hat{x}_i^{(1)}\sum_{e\in\mathcal{E}(i)}\sum_{j\in e}h_{i,j}^{e,-}(\bar{x}^{(2)}-\bar{x}^{(1)})+\frac{\delta}{N_1}\sum_{i=1}^{N_1}\hat{x}_i^{(1)}x_i^{(1)}(1-(x_i^{(1)})^2)$$

$$=:I_1+I_2+I_3+I_4.$$

$I_1$ can be defined by

$$I_1=\frac{1}{N_1}\sum_{i=1}^{N_1}\hat{x}_i^{(1)}\sum_{e\in\mathcal{E}(i)}\sum_{i'\in e}h_{i,i'}^{e,+}(\hat{x}_{i'}^{(1)}-\hat{x}_i^{(1)})$$

$$=\frac{1}{N_1}\sum_{e\in\mathcal{E}}\sum_{i,i'\in e}h_{i,i'}^{e,+}(\hat{x}_{i'}^{(1)}-\hat{x}_i^{(1)})\hat{x}_i^{(1)} \qquad (51)$$

$$=\frac{1}{N_1}\sum_{e\in\mathcal{E}}((\hat{x}^{(1)})^e)^\top A^e(\hat{x}^{(1)})^e,$$

where $(\hat{x}^{(1)})^e:=(\hat{x}_{i_1}^{(1)},\cdots,\hat{x}_{i_{|e|}}^{(1)})^\top$ and $A^e$ is given by Eq. 22 for each $e$. Thus $I_1$ is bounded by

$$\frac{1}{N_1}\sum_{e\in\mathcal{E}}((\hat{x}^{(1)})^e)^\top A^e(\hat{x}^{(1)})^e\leq-\frac{1}{N_1}\sum_{e\in\mathcal{E}}C^A\|(\hat{x}^{(1)})^e\|^2\leq-\frac{c_r}{N_1}\sum_{i=1}^{N_1}C^A\|\hat{x}_i^{(1)}\|^2, \qquad (52)$$

where $c_r$ is a constant larger than 1 related to the repetition of $\{\hat{x}_i^{(1)}\}$ in all hyperedges.

$I_2$ can be controlled by

$$I_2=-\frac{1}{N_1}\sum_{i=1}^{N_1}\hat{x}_i^{(1)}\sum_{e\in\mathcal{E}(i)}\sum_{j\in e}h_{i,j}^{e,-}(\hat{x}_j^{(2)}-\hat{x}_i^{(1)}) \qquad (53)$$

$$=-\frac{1}{N_1}\sum_{e\in\mathcal{E}}\sum_{i,j\in e}h_{i,j}^{e,-}(\hat{x}_j^{(2)}-\hat{x}_i^{(1)})\hat{x}_i^{(1)} \qquad (54)$$

$$=-\frac{1}{N_1}\sum_{e\in\mathcal{E}}\sum_{i,j\in e}h_{i,j}^{e,-}\hat{x}_j^{(2)}\hat{x}_i^{(1)}+\frac{1}{N_1}\sum_{e\in\mathcal{E}}\sum_{i,j\in e}h_{i,j}^{e,-}\|\hat{x}_i^{(1)}\|^2 \qquad (55)$$

$$\leq \frac{1}{N_1} \sum_{e \in \mathcal{E}} \sum_{i,j \in e} h_{i,j}^{e,-} \frac{1}{2} (\|\hat{x}_j^{(2)}\|^2 + \|\hat{x}_i^{(1)}\|^2) + \frac{1}{N_1} \sum_{e \in \mathcal{E}} D_e^- \sum_{i \in e} \|\hat{x}_i^{(1)}\|^2 \tag{56}$$

$$\leq \frac{1}{2N_1} \sum_{e \in \mathcal{E}} \sum_{j \in e} \psi_j^{e,-} \|\hat{x}_j^{(2)}\|^2 + \frac{3D^-}{2N_1} \sum_{e \in \mathcal{E}} \sum_{i \in e} \|\hat{x}_j^{(1)}\|^2 \tag{57}$$

$$\leq \frac{D^-}{2N_1} \sum_{e \in \mathcal{E}} \sum_{j \in e} \|\hat{x}_j^{(2)}\|^2 + \frac{3D^-}{2N_1} \sum_{e \in \mathcal{E}} \sum_{i \in e} \|\hat{x}_j^{(1)}\|^2 \tag{58}$$

$$\leq \frac{D^- k}{2N_1} \sum_{j=1}^{N_2} \|\hat{x}_j^{(2)}\|^2 + \frac{3D^- k}{2N_1} \sum_{i=1}^{N_1} \|\hat{x}_i^{(1)}\|^2. \tag{59}$$

$I_3$ has the below estimate for any constant $c_2 > 0$:

$$
\begin{aligned}
I_3 = & -\frac{1}{N_1} \sum_{i=1}^{N_1} \hat{x}_i^{(1)} \sum_{e \in \mathcal{E}(i)} \sum_{j \in e} h_{i,j}^{e,-} (\bar{x}^{(2)} - \bar{x}^{(1)}) \\
\leq & c_2 \|\bar{x}^{(1)} - \bar{x}^{(2)}\|^2 + \frac{1}{4c_2 N_1} \sum_{e \in \mathcal{E}} \sum_{i,j \in e} \|h_{i,j}^{e,-}\|^2 \|\hat{x}_i^{(1)}\|^2 \\
\leq & c_2 \|\bar{x}^{(1)} - \bar{x}^{(2)}\|^2 + \frac{(D^-)^2}{4c_2 N_1} \sum_{e \in \mathcal{E}} \sum_{i,j \in e} \|\hat{x}_i^{(1)}\|^2 \\
\leq & c_2 \|\bar{x}^{(1)} - \bar{x}^{(2)}\|^2 + \frac{(D^-)^2 k}{4c_2 N_1} \sum_{e \in \mathcal{E}} \sum_{i=1}^{N_1} \|\hat{x}_i^{(1)}\|^2.
\end{aligned}
\tag{60}
$$

Define

$$
\begin{aligned}
I_4 := & \frac{1}{N_1} \delta \sum_{i=1}^{N_1} \hat{x}_i^{(1)} x_i^{(1)} (1 - \|x_i^{(1)}\|^2) \\
= & \frac{\delta}{N_1} \sum_{i=1}^{N_1} \hat{x}_i^{(1)} (\hat{x}_i^{(1)} + \bar{x}^{(1)})(1 - (x_i^{(1)})^2) \\
= & \frac{\delta}{N_1} \sum_{i=1}^{N_1} (\hat{x}_i^{(1)})^2 - \frac{\delta}{N_1} \sum_{i=1}^{N_1} (\hat{x}_i^{(1)})^2 (x_i^{(1)})^2 + \frac{\delta}{N_1} \sum_{i=1}^{N_1} \hat{x}_i^{(1)} \bar{x}^{(1)} - \frac{\delta}{N_1} \sum_{i=1}^{N_1} \hat{x}_i^{(1)} (x_i^{(1)})^2 \bar{x}^{(1)} \\
= & \frac{\delta}{N_1} \sum_{i=1}^{N_1} (\hat{x}_i^{(1)})^2 - \frac{\delta}{N_1} \sum_{i=1}^{N_1} \hat{x}_i^{(1)} \|x_i^{(1)}\|^2 x_i^{(1)}.
\end{aligned}
\tag{61}
$$

Note that

$$
\begin{aligned}
\sum_{i=1}^{N_1} \hat{x}_i^{(1)} \|x_i^{(1)}\|^2 x_i^{(1)} = & \sum_{i=1}^{N_1} \|x_i^{(1)}\|^2 (\|x_i^{(1)}\|^2 - x_i^{(1)} \bar{x}^{(1)}) \\
\geq & \frac{1}{2} \sum_{i=1}^{N_1} \|x_i^{(1)}\|^2 (\|x_i^{(1)}\|^2 - \|\bar{x}^{(1)}\|^2) \\
= & \frac{1}{2} \sum_{i=1}^{N_1} \|x_i^{(1)}\|^4 - \frac{1}{2} \sum_{i=1}^{N_1} \|x_i^{(1)}\|^2 \|\bar{x}^{(1)}\|^2 \\
\geq & \frac{1}{2} \sum_{i=1}^{N_1} \|x_i^{(1)}\|^4 - \frac{1}{2N_1} \sum_{i=1}^{N_1} (\|x_i^{(1)}\|^2)^2 \\
\geq & 0.
\end{aligned}
\tag{62}
$$

Hence,

$$I_4 \leq \delta \widehat{M}_2(u). \tag{63}$$

Then

$$\frac{\mathrm{d}}{\mathrm{d}t}\left(\frac{1}{2N_1}\sum_{i=1}^{N_1}\|\hat{x}_i^{(1)}\|^2\right)$$

$$\leq k\left(-\frac{C^A}{N_1}+\frac{3D^-}{2N_1}+\frac{(D^-)^2}{4c_2N_1}\right)\sum_{i=1}^{N_1}\|\hat{x}_i^{(1)}\|^2+k\frac{D^-}{2N_2}\sum_{j=1}^{N_2}\|\hat{x}_j^{(2)}\|^2+c_2\|\bar{x}^{(1)}-\bar{x}^{(2)}\|^2 \quad (64)$$

$$\leq\left[k\left(-C^A+\frac{3D^-}{2}+\frac{(D^-)^2}{4c_2}\right)+\delta\right]\widehat{M_2}(u)+k\frac{D^-}{2}\widehat{M_2}(v)+c_2\|\bar{x}^{(1)}-\bar{x}^{(2)}\|^2.$$

Similarly,

$$\frac{\mathrm{d}}{\mathrm{d}t}\left(\frac{1}{2N_2}\sum_{j=1}^{N_2}\|\hat{x}_j^{(2)}\|^2\right)$$

$$\leq\left[k\left(-C^A+\frac{3D^-}{2}+\frac{(D^-)^2}{4c_2}\right)+\delta\right]\widehat{M_2}(v)+k\frac{D^-}{2}\widehat{M_2}(u)+c_2\|\bar{x}^{(1)}-\bar{x}^{(2)}\|^2. \tag{65}$$

Summing them together gives

$$\frac{1}{2}\frac{\mathrm{d}}{\mathrm{d}t}(\widehat{M_2})\leq -k\left(C^A-\frac{D^-}{2}+\frac{(D^-)^2}{4c_2}-\frac{\delta}{k}\right)(\widehat{M_2})+2c_2\|\bar{x}^{(1)}-\bar{x}^{(2)}\|^2. \tag{66}$$

This gives an exponential growth estimate or $\widehat{M_2}$ up to an error term of $\|\bar{x}^{(1)}-\bar{x}^{(2)}\|^2$. ∎

**Proposition B.5** ($L_2$ separation of HAMP-I). *For Eq. 4, suppose the above assumptions are satisfied. Define the mean value $\bar{\mathbf{x}}:=\frac{1}{N}\sum_{i=1}^{N}\mathbf{x}_i$, and the second moments $M_2(\mathbf{x}):=\sum_{i=1}^{N}\mathbf{x}_i^2$. Then for sufficiently large $N_1, N_2$, there exist constants $\lambda_-, \lambda_+$, such that if the initial data satisfies*

$$\lambda(0):=\frac{\widehat{M_2}(0)}{\|\bar{\mathbf{x}}^{(1)}(0)-\bar{\mathbf{x}}^{(2)}(0)\|^2}\leq\lambda_+, \tag{67}$$

*then, there holds that the $L_2$ separation*

$$\lambda(t):=\frac{\widehat{M_2}(t)}{\|\bar{\mathbf{x}}^{(1)}(t)-\bar{\mathbf{x}}^{(2)}(t)\|^2}\leq\lambda_-+(\lambda(0)-\lambda_-)e^{-\mu t}, \tag{68}$$

*with a positive constant $\mu$, where $\widehat{M_2}(t):=M_2(\mathbf{x}^{(1)}(t))+M_2(\mathbf{x}^{(2)}(t))$.*

*Proof of Proposition 5.2.* Lemma B.3 gives an exponential growth estimate of $\|\bar{x}^{(1)}-\bar{x}^{(2)}\|^2$ up to an error term of $\widehat{M_2}$. If $N_0$ is large enough, the coefficient of the error term is small.

Lemma B.4 gives an exponential growth estimate or $\widehat{M_2}$ up to an error term of $\|\bar{x}^{(1)}-\bar{x}^{(2)}\|^2$. If $N_0$ is large enough, the coefficient of the error term is small.

Set

$$A_{11}:=\frac{2c_m}{N_0}-c_1,\ A_{12}:=\frac{4(D_2^-)^2 k}{c_1 N_0},\ A_{21}=2c_2,\ A_{22}=k\left(C^A-\frac{D^-}{2}+\frac{(D^-)^2}{4c_2}\right). \tag{69}$$

If $N_0$ is large enough, Eq. 69 is satisfied.

$$(A_{11}+A_{12})^2-4A_{21}A_{12}>0. \tag{70}$$

Apply Lemma 4.1 in [27] then we obtain the $L_2$ separation. ∎

**Remark B.6.** We can alternatively prove Theorem 5.2 by the method of [[44] Proposition 2 and 3].

**Proposition B.7** (Lower bound of the Dirichlet energy). *If the hypergraph $\mathcal{H}$ is a connected one, for Eq. 4 with the conditions of Theorem 5.2, or for Eq. 5 with conditions of Theorem 5.1 in [19], there exists a positive lower bound of the Dirichlet energy.*

*Proof of Proposition 5.4.* The relative size between $\|\bar{x}^{(1)} - \bar{x}^{(2)}\|^2$ and $\widehat{M_2}$ is an indicator of group separation in the sense of $L_2$: if $\|\bar{x}^{(1)} - \bar{x}^{(2)}\|^2$ is much larger than $\widehat{M_2}$, then the two groups are well-separated in average sense. Since the hypergraph is connected, there is a positive bound between different clusters, hence the Dirichlet energy does not decay to zero.

**Remark B.8.** The proof for the second-order system Eq. 5 can be proved in a similar way.

**Remark B.9.** The separability of Eq. 6 which is the Eq. 4 or Eq. 5 with a stochastic term also holds.

## C   Experiment Details

### C.1   Dataset Details

We selected the most common hypergraph benchmark datasets, and the statistics of nine datasets across different domains are summarized in Tab. 3. The key statistics include the number of nodes, hyperedges, features, classes, average node degree $d_v$, average hyperedge size $|\mathcal{E}|$, and CE homophily [36]. These variations highlight the diversity in dataset scales and structural patterns, which may influence model performance in hypergraph-related tasks.

Table 3: The summary of data statistics.

| Dataset | # nodes | # hyperedges | # features | # classes | avg. $d_v$ | avg. $|\mathcal{E}|$ | CE homophily |
|---------|---------|--------------|------------|-----------|-----------|---------------------|--------------|
| Cora | 2708 | 1579 | 1433 | 7 | 1.767 | 3.03 | 0.897 |
| Citeseer | 3312 | 1079 | 3703 | 6 | 1.044 | 3.200 | 0.893 |
| Pubmed | 19717 | 7963 | 500 | 3 | 1.756 | 4.349 | 0.952 |
| Cora-CA | 2708 | 1072 | 1433 | 7 | 1.693 | 4.277 | 0.803 |
| DBLP-CA | 41302 | 22363 | 1425 | 6 | 2.411 | 4.452 | 0.869 |
| Congress | 1718 | 83105 | 100 | 2 | 427.237 | 8.656 | 0.555 |
| House | 1290 | 340 | 100 | 2 | 9.181 | 34.730 | 0.509 |
| Senate | 282 | 315 | 100 | 2 | 19.177 | 17.168 | 0.498 |
| Walmart | 88860 | 69906 | 100 | 11 | 5.184 | 6.589 | 0.530 |

### C.2   Additional Ablation Studies

**Impact of Hidden Dimension on HAMP-I and HAMP-II.**   We explore the tolerance of our model to different hidden dimensions, as shown in Tab. 4. For simplicity, we only vary the size of hidden dimension, while other parameters remain fixed. Overall, these results demonstrate the robustness of our methods with varying hidden dimensions. It is worth noting that high hidden dimension is key to achieving the best performance of HAMP.

Table 4: Impact of hidden dimension evaluated on the hypergraph datasets.

| | HAMP-I | | | | HAMP-II | | | |
|---|-------|------|-------|-------|--------|-------|-------|-------|
| | 128 | 256 | 512 | 1024 | 128 | 256 | 512 | 1024 |
| Cora | 80.19 | 80.55 | **81.18** | 79.85 | 79.59 | 79.54 | **80.80** | 79.60 |
| Citeseer | 73.39 | 72.89 | **75.22** | 72.40 | 73.95 | 74.36 | **75.33** | 75.19 |
| Pubmed | 88.49 | 88.85 | **89.02** | 88.82 | 88.48 | 88.48 | **89.05** | 88.78 |
| Senate | 63.24 | 65.35 | **69.44** | 66.62 | 63.52 | 63.10 | **70.14** | 62.96 |
| House | 71.21 | 70.62 | 72.66 | **72.72** | 68.30 | 69.47 | **72.60** | 71.21 |

**Impact of Repulsion, Allen-Cahn Force, and Noise on HAMP-I and HAMP-II.**   We summary ablation studies to investigate the individual and combined effects of repulsion $f_\beta^-$, Allen-Cahn force $f_d$, and noise $B_t$ on both HAMP-I and HAMP-II. Tab. 5 reports the average node classification accuracy with a standard deviation across seven standard hypergraph benchmarks over 10 runs. Models with the repulsion term enabled outperform their counterparts in some dataset, indicating

enhanced ability to distinguish node representations in complex hypergraph structures. The synergy between the repulsion and Allen-Cahn terms further boosts performance, confirming that these particle system-inspired mechanisms play complementary roles. Overall, these improvement confirm the validity of the HAMP construction and further highlight the significant advantages of incorporating particle system theory into the hypergraph message passing learning process.

Table 5: Ablation studies on some standard hypergraph benchmarks. The accuracy (%) is reported with a standard deviation from 10 repetitive runs. (Key: $f_\beta^-$: repulsion; $f_d$: Allen-Cahn; $B_t$: noise.)

| Homophilic | $f_\beta^-$ | $f_d$ | $B_t$ | Cora | Citeseer | Pubmed | Cora-CA |
|---|---|---|---|---|---|---|---|
| | ✗ | ✗ | ✗ | 76.09±1.22 | 70.53±1.56 | 87.98±0.38 | 83.13±1.26 |
| | ✓ | ✗ | ✗ | 76.40±1.56 | 70.85±1.65 | 88.25±0.50 | 83.15±1.36 |
| | ✗ | ✓ | ✗ | 80.31±1.41 | 74.83±1.70 | 88.90±0.45 | 84.77±1.16 |
| HAMP-I | ✗ | ✗ | ✓ | 75.67±1.71 | 70.59±1.40 | 87.93±0.52 | 82.70±1.01 |
| | ✓ | ✓ | ✗ | 80.49±1.26 | 74.96±1.56 | 88.87±0.40 | 85.21±1.49 |
| | ✗ | ✓ | ✓ | 80.59±1.25 | 74.67±1.69 | 88.77±0.44 | 84.59±1.03 |
| | ✓ | ✓ | ✓ | **81.18±1.30** | **75.22±1.62** | **89.02±0.49** | **85.23±1.15** |
| | ✗ | ✗ | ✗ | 77.42±1.44 | 71.50±1.49 | 88.68±0.62 | 83.60±1.45 |
| | ✓ | ✗ | ✗ | 77.08±1.73 | 72.20±1.14 | 88.59±0.50 | 82.95±1.35 |
| | ✗ | ✓ | ✗ | 80.13±1.26 | 74.37±1.59 | 88.86±0.55 | 84.37±1.45 |
| HAMP-II | ✗ | ✗ | ✓ | 77.18±1.61 | 71.75±1.68 | 88.68±0.59 | 82.91±1.28 |
| | ✓ | ✓ | ✗ | 79.70±1.36 | 73.99±1.75 | 88.82±0.48 | 84.30±1.32 |
| | ✗ | ✓ | ✓ | 79.50±1.25 | 74.25±1.28 | 88.80±0.40 | 83.31±1.44 |
| | ✓ | ✓ | ✓ | **80.80±1.62** | **75.33±1.61** | **89.05±0.41** | **84.89±1.14** |

Table 6: Node Classification on standard hypergraph benchmarks. The accuracy (%) is reported with a standard deviation from 10 repetitive runs. (Key: $f_\beta^-$: repulsion; $f_d$: Allen-Cahn; $B_t$: noise.)

| Heterophilic | $f_\beta^-$ | $f_d$ | $B_t$ | Congress | Senate | Walmart | House |
|---|---|---|---|---|---|---|---|
| | ✗ | ✗ | ✗ | 93.51±1.08 | 60.70±8.38 | 69.64±0.35 | 70.50±1.45 |
| | ✓ | ✗ | ✗ | 93.70±1.02 | 60.00±8.66 | 69.80±0.45 | 70.56±1.96 |
| | ✗ | ✓ | ✗ | 94.79±1.14 | 66.76±5.44 | 69.77±0.28 | 71.55±1.53 |
| HAMP-I | ✗ | ✗ | ✓ | 93.47±1.13 | 60.14±7.54 | 69.86±0.35 | 69.88±2.48 |
| | ✓ | ✓ | ✗ | 94.58±1.25 | 67.75±8.82 | 69.76±0.37 | 71.58±1.87 |
| | ✗ | ✓ | ✓ | 94.67±1.02 | 65.63±2.98 | 69.73±0.49 | 71.55±2.58 |
| | ✓ | ✓ | ✓ | **95.09±0.79** | **69.44±6.09** | **69.90±0.38** | **72.72±1.77** |
| | ✗ | ✗ | ✗ | 94.79±0.73 | 58.73±7.03 | 69.84±0.25 | 69.85±1.61 |
| | ✓ | ✗ | ✗ | 94.02±1.10 | 61.55±5.98 | 69.91±0.30 | 69.35±1.87 |
| | ✗ | ✓ | ✗ | 94.19±1.07 | 62.82±6.44 | 69.89±0.31 | 70.96±2.06 |
| HAMP-II | ✗ | ✗ | ✓ | 94.35±1.14 | 60.14±5.10 | 69.84±0.37 | 70.46±2.08 |
| | ✓ | ✓ | ✗ | 94.58±0.86 | 61.97±8.42 | 69.92±0.33 | 71.36±1.68 |
| | ✗ | ✓ | ✓ | 94.12±0.63 | 64.51±4.19 | 69.86±0.34 | 70.96±2.76 |
| | ✓ | ✓ | ✓ | **95.26±1.34** | **70.14±6.08** | **69.94±0.37** | **72.60±1.23** |

## C.3 Time-Memory Tradeoff Analysis

We intuitively reveal the differences of different methods with a single-layer network in the time-memory trade-off on Walmart dataset. As shown in Fig. 5, HAMP-I and HAMP-II methods demonstrate a notable trade-off between time efficiency and memory consumption. The experimental results reveal:

- Memory usage: HAMP-I and HAMP-II maintain memory consumption within 11000-12000 MiB, achieving a 20-26% reduction compared to ED-HNN, while being comparable to HDS$^{ode}$.

- Time efficiency: Although the time consumption for HAMP-I and HAMP-II runtime slightly exceeds ED-HNN (0.1s), it outperforms the baseline HDS$^{ode}$.

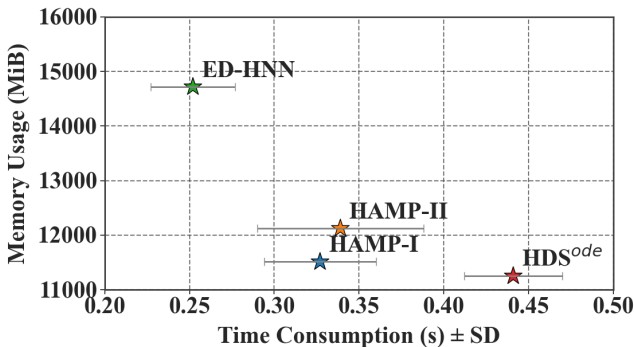

Figure 5: Time-Memory tradeoff analysis of different methods on Walmart dataset. SD denotes the standard deviation of time consumption.

## C.4 Hyperparameters

To ensure fairness, we follow the same training recipe as ED-HNN. Specifically, we train the model for 500 epochs using the Adam optimizer with the learning rate of 0.001 and no weight decay during the training phases. And we apply early stopping with a patience of 50. For the stability, we run 10 trials with different seed and report the results of mean and the standard deviation. All experiments are implemented on an NVIDIA RTX 4090 GPU with Pytorch.

We explore the parameter space by grid search, where the search ranges for each critical hyperparameter are delineated below:

- Dropout rate in {0.1, 0.2, 0.3, 0.4, 0.5, 0.6, 0.7, 0.8, 0.9};
- Layer number of classifier in {1, 2, 3};
- The hidden dimension of classifier in {128, 256, 512};
- Hidden dimension of model in {128, 256, 512, 1024};
- step size of solver in {0.09, 0.1, 0.15, 0.2, 0.25};
- $\gamma$ of repulsive force in {0.01, 0.02, 0.03, 0.04, 0.05, 0.06, 0.07, 0.08, 0.09, 0.1, 0.11, 0.12, 0.13, 0.14, 0.15};
- Initial values of learnable parameters $\delta$ of damping term in {0, 1, 2, 3, 4, 5, 6, 7, 8, 9, 10, 11, 12, 13, 14, 15};
- Initial values of learnable parameters $\epsilon$ of noise term in {0, 0.1, 0.3};

Tab. 7 and Tab. 8 summarize the best hyperparameters on standard hypergraph benchmarks using HAMP-I and HAMP-II, respectively. For fairness, a linear layer is added to perform feature mapping when conducting $HDS^{ode}$ experiments. The optimal hyperparameters for node classification on standard hypergraph benchmarks is achieved by the $HDS^{ode}$ algorithm, as demonstrated in Tab. 9.

Table 7: The best hyperparameters of Node Classification on standard hypergraph benchmarks using the HAMP-I algorithm.

| Dataset | model. hd | cls. hd and # layers | time | step size | $\delta$ | $\gamma$ | dropout | $\epsilon$ |
|---------|-----------|---------------------|------|-----------|----------|----------|---------|-----------|
| Cora | 512 | 128, 1 | 1 | 0.1 | 12 | 0.05 | 0.4 | 0 |
| Citeseer | 512 | 512, 1 | 0.6 | 0.1 | 6 | 0.05 | 0.2 | 0 |
| Pubmed | 512 | 256, 1 | 0.2 | 0.1 | 15 | 0.1 | 0.5 | 0 |
| Cora-CA | 512 | 512, 2 | 0.4 | 0.2 | 4 | 0.05 | 0.9 | 0 |
| DBLP-CA | 256 | 128, 2 | 1.1 | 0.1 | 11 | 0.12 | 0.2 | 0 |
| Congress | 128 | 128, 2 | 1.4 | 0.1 | 1 | 0.08 | 0.3 | 0 |
| House | 1024 | 512, 3 | 1.05 | 0.15 | 3 | 0.05 | 0.8 | 0 |
| Senate | 512 | 256, 2 | 0.6 | 0.1 | 10 | 0.05 | 0.7 | 0 |
| Walmart | 256 | 128, 2 | 1.75 | 0.25 | 0 | 0.02 | 0.3 | 0 |

Table 8: The best hyperparameters of Node Classification on standard hypergraph benchmarks using the HAMP-II algorithm.

| Dataset | model. hd | cls. hd and layers | time | step size | $\delta$ | $\gamma$ | dropout | $\epsilon$ |
|---------|-----------|-------------------|------|-----------|----------|----------|---------|-----------|
| Cora | 512 | 512, 1 | 1.9 | 0.1 | 5 | 0.12 | 0.3 | 0.1 |
| Citeseer | 512 | 512, 1 | 1.8 | 0.15 | 8 | 0.13 | 0.6 | 0 |
| Pubmed | 512 | 256, 1 | 0.6 | 0.09 | 5 | 0.09 | 0.3 | 0 |
| Cora-CA | 512 | 128, 2 | 0.75 | 0.25 | 3 | 0.01 | 0.7 | 0 |
| DBLP-CA | 256 | 128, 2 | 3.45 | 0.15 | 7 | 0.09 | 0.3 | 0 |
| Congress | 128 | 128, 2 | 6.25 | 0.25 | 0 | 0.01 | 0.3 | 0 |
| House | 512 | 256, 2 | 1.6 | 0.1 | 8 | 0.14 | 0.8 | 0 |
| Senate | 512 | 256, 2 | 3 | 0.2 | 13 | 0.05 | 0.3 | 0.3 |
| Walmart | 256 | 128, 2 | 2.5 | 0.25 | 0 | 0.02 | 0.3 | 0 |

Table 9: The best hyperparameters of Node Classification on standard hypergraph benchmarks using the HDS$^{ode}$ algorithm.

| Dataset | model. hd | cls. hd and layers | # layer model. | alpha$_v$ | alpha$_e$ | step |
|---------|-----------|-------------------|----------------|-----------|-----------|------|
| Cora | 512 | 256, 2 | 10 | 0.05 | 0.9 | 20 |
| Citeseer | 512 | 512, 1 | 5 | 0.05 | 0.9 | 20 |
| Pubmed | 512 | 256, 1 | 12 | 0.05 | 0.9 | 20 |
| Cora-CA | 512 | 512, 2 | 9 | 0.05 | 0.9 | 20 |
| DBLP-CA | 256 | 256, 2 | 15 | 0.05 | 0.9 | 20 |
| Congress | 256 | 128, 2 | 7 | 0.25 | 0.9 | 5 |
| House | 512 | 256, 2 | 10 | 0.05 | 0.9 | 20 |
| Senate | 512 | 256, 2 | 9 | 0.05 | 0.9 | 20 |
| Walmart | 256 | 128, 2 | 6 | 0.25 | 0.9 | 5 |

