# OpenReview forum: "How Particle System Theory Enhances Hypergraph Message Passing"
_NeurIPS.cc/2025/Conference — NeurIPS 2025 poster_

### Official Review · Reviewer_L9c7 · 2025-07-02

**Clarity:** 3
**Significance:** 3
**Originality:** 3
**Rating:** 4
**Confidence:** 4

**Summary:**

This work proposes a novel hypergraph message passing framework inspired by particle system theory. The work suggests two models for message passing for HNNs based on first-order and second-order system equations. The suggested models are proved to mitigate over-something and empirically shown to produce good results on the tested benchmarks.

**Questions:**

What is bi-flocking? (line 200)

**Ethical Concerns:**

["NO or VERY MINOR ethics concerns only"]

**Final Justification:**

Since the rebuttal did not include any new experiments involving additional datasets, and the reviewers did not clearly specify how the new clarifications would be incorporated into the main text, I have decided to maintain my original score.

**Limitations:**

yes

**Quality:**

3

**Strengths And Weaknesses:**

Strengths

The suggested method has a theoretical background which ensures that it avoids over-smoothing

An ablation study was conducted, showing that both repulsion and Allen-Cahn improve Node Classification results.

The suggested method generally performs better than the compared baselines

Weaknesses

As far as I know, Cora, Citeseer and Pubmed are not Hypergraph datasets but graph datasets.

The improvement of HAMP is insignificant with respect to the second performing baseline.

From the results, HAMP-|| indeed outperforms the other baselines when using deep layers, but its performance converges very fast, even when the network is shallow, so there is no extra advantage in using deep HAMP-||?

It would be interesting to see the performance of the model on particle system datasets.

---

> ### Author Rebuttal · Authors · 2025-07-30
>
> **Q1: Dataset Type (Cora, Citeseer, Pubmed)**.
>
> We appreciate the reviewer's vigilance regarding data types. The reviewer is correct that Cora, Citeseer, and Pubmed were originally introduced as graph datasets. However, in hypergraph representation learning research, it is standard practice to convert these citation networks into hypergraphs to capture higher-order relationships. The key transformation lies in defining the hyperedges. Specifically, following the methodology of ED-HNN [1] and common practice in the field, for co-citation networks (Cora, Citeseer, Pubmed), all documents cited by a given document are connected by a hyperedge. As previously stated, ED-HNN, which our work builds upon, explicitly follows this established methodology. This conversion allows for the evaluation of hypergraph models on established benchmarks.
>
> **Q2: Performance improvements.**
>
> Our model, HAMP, demonstrates superior performance through consistent gains and stability across key benchmarks. As detailed in Table 1, HAMP achieves strong results on all nine datasets, including a notable 3% improvement on both the Senate and Walmart datasets.
>
> More importantly, HAMP-II shows exceptional robustness as network depth increases. Figure 2 illustrates that while competing methods suffer from significant performance degradation at greater depths, HAMP-II consistently outperforms them. This highlights HAMP's ability to capture complex representations while maintaining stability, establishing it as a valuable and reliable framework for deep Hypergraph Neural Networks (HNNs).
>
> **Q3: Extra advantage of using deep HAMP-II.**
>
> We will clarify HAMP-II's depth behavior from both theoretical and empirical perspectives. As our theoretical analysis in the article shows, based on the particle system perspective, HAMP enables stable information propagation through deeper layers. From an empirical standpoint, HAMP-II achieves rapid convergence on locally structured datasets (e.g., Cora, CiteSeer) with shallow architectures (2-3 layers), demonstrating parameter efficiency. However, for datasets exhibiting long-range dependencies (e.g., Congress, Walmart), deep architectures are essential to capture multi-hop interactions. As depicted in Figure 2, on the Congress dataset, a 10-layer HAMP-II model improves accuracy over its 2-layer counterpart, whereas ED-HNN degrades due to over-smoothing. This demonstrates HAMP-II's unique capacity to leverage depth for complex topologies without performance collapse.
>
> **Q4: The performance of the model on particle system datasets.**
>
> Exploring HAMP within the context of particle systems is indeed an interesting direction, as the continuous dynamics of such systems naturally align with our HAMP framework. Our current evaluation, however, focuses on discrete hypergraph benchmarks to facilitate direct comparison with the existing hypergraph neural network literature.
>
> **Q5: Definition of bi-flocking.**
> Thank you for this insightful question, it has revealed an important conceptual inaccuracy in our original terminology. This prompted us to replace all instances of ''bi-flocking'' with the more precise term ''bi-cluster flocking'' adopted from [2].
> Next, we will elaborate on the concept of bi-cluster flocking, which is characterized by the following two conditions:
> - *Intra-cluster cohesion*: bounded position differences and velocity alignment within each sub-ensemble.
> - *Inter-cluster separation*: diverging positions between sub-ensembles.
>
>
> **References:**
>
> [1] Wang, et al. Equivariant hypergraph diffusion neural operators. In *ICLR*, 2023.
>
> [2] Fang, et al. Emergent behaviors of the Cucker-Smale ensemble under attractive-repulsive couplings and rayleigh frictions. *Mathematical Models and Methods in Applied Sciences*, 29(07):1349–1385, 2019.

---

> > ### Comment · Reviewer_L9c7 · 2025-08-03
> > **Response to Rebuttal**
> >
> > Since the rebuttal did not include any new experiments involving additional datasets, and the reviewers did not clearly specify how the new clarifications would be incorporated into the main text, I have decided to maintain my original score.

---

> > > ### Author Response · Authors · 2025-08-05
> > >
> > > Thank you for your message and for your continued engagement with our manuscript. We appreciate you reiterating your assessment and confirming the score. While we had hoped to address your concerns sufficiently to potentially raise the score, we fully respect your judgment and expertise. We are grateful for the detailed feedback you provided initially; it has been invaluable, and we will diligently incorporate your suggestions where appropriate to strengthen the paper for publication.

---

### Official Review · Reviewer_5ffs · 2025-07-02

**Clarity:** 3
**Significance:** 3
**Originality:** 3
**Rating:** 4
**Confidence:** 3

**Summary:**

This paper introduces Hypergraph Atomic Message Passing (HAMP), a framework inspired by particle systems, to address higher-order relationships in hypergraphs. Utilizing concepts like attraction-repulsion forces and Allen-Cahn potential dynamics, HAMP improves traditional hypergraph neural networks by modeling hyperedges as fields influencing particle-level node interactions. Theoretically, HAMP maintains a positive lower bound on hypergraph Dirichlet energy, ensuring deeper message passing without sacrificing efficiency.

**Questions:**

1. How does HAMP perform computationally on hypergraphs with millions of nodes and edges? Are there optimizations to alleviate complexity at scale?
2. HDS^ode also models the hypergraph dynamic system, what are the main advantages of HAMP being more effective compared to it?

**Ethical Concerns:**

["NO or VERY MINOR ethics concerns only"]

**Final Justification:**

Comprehensively considering the original paper's innovation and contribution, and the authors' rebuttal, I keep my score (Borderline accept).

**Limitations:**

yes

**Quality:**

3

**Strengths And Weaknesses:**

- Strengths:
1. HAMP leverages particle system theory to model interactions, bridging physics concepts with hypergraph learning for deeper insights.
2. The integration of repulsion and Allen-Cahn dynamics effectively reduces over-smoothing, enabling deeper architectures.
3. Experimental results show HAMP achieves superior accuracy, particularly on challenging heterophilic hypergraph benchmarks.
4. The code is available.

- Weaknesses:
1. The use of first-order and second-order systems adds computational overhead, potentially limiting scalability to larger hypergraphs.
2. While effective for classification, broader application scenarios (e.g., graph classification) remain unexplored.

---

> ### Author Rebuttal · Authors · 2025-07-30
>
> We really appreciate you for the valuable insights. All concerns are addressed below point by point.
>
> **Q1: Scalability of HAMP.**
>
> As mentioned in the appendix A and C.3, we analyze the computational complexity of one layer in HAMP-I/II. This aligns with efficient ED-HNN. We show Time-Memory tradeoff analysis of different methods on Walmart dataset ($|\mathcal{V}| \approx 88k$, $|\mathcal{E}| \approx  69k$). Results (Figure 5) show: HAMP-I/II use 20–26% less memory than ED-HNN. Runtime is near-identical to ED-HNN ($\Delta t \approx 0.1$s) and faster than HDS$^{ode}$.
>
> **Q2: Broader application scenarios.**
>
> Our current experiments focus on hypergraph node classification to ensure a fair comparison with state-of-the-art methods like ED-HNN and AllDeepSets. This approach also serves to validate our core message-passing mechanism on established hypergraph benchmarks. While this work centers on hypergraph node classification, we recognize the potential of HAMP for other learning tasks. Future work will explore extending its versatility to hyperedge prediction, implemented by scoring pairwise node representations, and to hypergraph classification, using a global pooling mechanism. These extensions will further demonstrate the robustness and adaptability of the HAMP framework.
>
> **Q3: The main advantages of HAMP compared to HDS$^{ode}$.**
>
> As discussed in our related work, HDS$^{ode}$ adopts control-diffusion ODEs to model hypergraph dynamic systems. In contrast, we design hypergraph message passing from the perspective of particle system theory, which inherently alleviates the over-smoothing problem. Previous methods predominantly neglect a key physical insight: **message passing is inherently a particle-level phenomenon, involving local, interactive dynamics that cannot be fully captured by pure diffusion models**. Our core innovation lies in modeling hyperedges as dynamic interaction fields between node-level particles, where class-dependent equilibrium distributions in the feature space are achieved through particle dynamics equations incorporating attractive-repulsive forces and Allen-Cahn potential regularization.

---

### Official Review · Reviewer_nNPk · 2025-07-02

**Clarity:** 2
**Significance:** 2
**Originality:** 2
**Rating:** 2
**Confidence:** 4

**Summary:**

This paper presents a framework for hypergraph message passing based on the perspective that the nodes are “particles” with features living in some embedding space, and these features are updated by pairwise “forces” between them mediated by the hyperedges.

They highlight the importance of having both attractive and repulsive terms to prevent over-smoothing, and include as well a global damping term for stability of the method and separation of the classes.

Their methods, HAMP-I and HAMP-II, update the node/particle features according to a first- and second-order differential equation, and they perform numerical studies demonstrating the effect of the damping terms on their performance.

**Questions:**

How is the repulsive force $f_\beta^-$ implemented?

The text referencing Table 2 seems to be off.  I see a larger increase from including Allen-Cahn $f_d$ than from the repulsive force $f_\beta^-$.

Is the double-well potential needed?  Could one get away with a potential that just grows sufficiently quickly so as to maintain stability?

**Ethical Concerns:**

["NO or VERY MINOR ethics concerns only"]

**Final Justification:**

## Outstanding concerns:

### **The significance of the results is questionable.**

Many of the numbers are in bold due to an accuracy difference of a fraction of a percent, and nearly all differences are less than the reported standard deviation.  As the only notable improvements are on the heterophilic datasets, I would need to know **how** the authors decided which nodes repel each other in order to assess the fairness of the comparison.  I asked this twice, and their responses did not leave me feeling I had a satisfactory answer.  The Class-Edge homophily metric requires knowing the class labels, and I still don't know how this translates to the magnitude of repulsion.  Is there just a single parameter controling this?  Is it tuned by hand to get higher accuracy?

*(First rebuttal to me, Paragraph "Q2: How to implement the repulsive force.")*
> "[...], we introduce a repulsive force. This force acts as a predefined prior to prevent homogenization---the uniform averaging of features across a hyperedge---by leveraging dataset-specific information. While effective, we note that a fully data-driven approach to modeling this repulsive force could further enhance the model's expressiveness, and we leave this for future work."

*(Second rebuttal to me, Section "2. Heterophily via repulsion")*
> "Ideally, repulsion serves to maintain the separation of nodes. However, in practical applications, precisely identifying repelling node pairs remains a challenge. Consequently, in instances where an attractive force exists between nodes, we implement a reduction in the magnitude of this attraction to implicitly account for repulsion. While this approach is a rudimentary implementation, the predefined reduction, leveraging dataset-specific information, effectively counteracts homogenization. We now elucidate the nature of this specific information.
>
> The Class-Edge homophily metric quantifies the consistency of class labels among nodes within the same hyperedge: $H_{CE} = \frac{1}{|E|} \sum_{e \in E} \frac{ \max_{c \in C} | { v \in e \mid y_v = c } | }{ |e| }.$ Here, ${E}$ is the set of hyperedges, ${C}$ is the set of node classes, and $y_v$ is the class label for node $v$. A higher $H_{CE}$ score indicates that hyperedges tend to connect nodes of the same class.
>
> For heterophilic datasets, such as Senate, where hyperedges predominantly connect dissimilar nodes, strong repulsion is indispensable for enhancing model performance. Conversely, homophilic datasets necessitate minimal repulsive interactions, thereby diminishing the utility of explicit repulsion mechanisms. Future research will explore data-driven methodologies to further refine the implementation of repulsion, aiming to achieve superior model performance."


### **The lower bound on the Dirichlet energy is not particularly useful.**
Their proof for the lower bound on the Dirichlet energy assumes *very* fine-tuned conditions.  In particular, my issue is with the assumption that the nodes can be partitioned into two groups, with no attraction between the two groups, and no repulsion within a group.  This is not just a technical condition (e.g., mild regularity conditions, or excluding a set of measure zero) --- it seems very unlikely to hold in practice, and seems to require that perfect node classification has already been achieved with zero outliers.  I am not convinced that this result constitutes a "theoretical guarantee" about their model's dynamics.

*(Second rebuttal to me, Section "1. Theoretical Interpretation")*
> "The incorporation of repulsive forces, provided certain conditions are met, endows the Dirichlet energy with a lower bound, thereby mitigating over-smoothing."


### **Text does not match figures.**
Figure 2: Allen-Cahn seems to help more than pairwise repulsion.  Text says the other way around.
Figure 3: Including noise doesn't seem to change much (is the legend correct?).  Text suggests it does.

On this note, they provided a table of accuracies with and without noise in (Second rebuttal to me, Section "3. Impact of noise on HAMP").  However, like the other numerical results about accuracies, when framed as a z-score, the results are rather lukewarm.


### **Final Justification**
While I appreciate the physics-inspired model for hypergraph message-passing, I still have some issues with the way they have presented their particular implementation.  Given these outstanding concerns, I would like to keep my current score.
However, if there is significant desire from the other reviewers, I would not want to stand in the way of publication.

**Limitations:**

No other limitations aside from the concerns/questions listed above.

**Quality:**

2

**Strengths And Weaknesses:**

Overall, the paper is written reasonably well.  And while I appreciate the physics-based perspective of particles interacting via forces, I am uncertain as to how the proposed framework is fundamentally different than message passing.  Equations 4 and 5 are special cases of message passing.

Propositions 5.2 and 5.3 already assume that within group $I_1$ or $I_2$ forces are attractive and between groups $I_1$ and $I_2$ forces are repulsive.  Though this seems unlikely in practice, if this were the case, then the partitioning problem is already solved.

It is unclear to me how the repulsive force for $f_\beta^-$ is implemented.

It would be good to have error bars for the “significance plot” in Figure 3.

---

> ### Author Rebuttal · Authors · 2025-07-30
>
> We appreciate your recognition of the writing and motivation of our work. Please find our detailed responses to your comments below.
>
> **Q1: Fundamental differentiation from classical message passing.**
>
> While Equations 4 and 5 can be expressed in a message passing form, the key novelty of HAMP lies in introducing a particle system perspective to message passing: we view message passing as the result of a physically inspired interacting particle system. This perspective enables several conceptual and practical advances beyond standard message passing schemes.
>
> First, traditional message passing typically aggregates information based on local neighborhood similarity and implicitly assumes homophily. In contrast, the proposed particle system formulation explicitly models both attractive and repulsive interactions between nodes, allowing it to handle heterophilic structures where neighbors may belong to different classes. The repulsive force mechanism is especially effective in promoting class separation, as demonstrated in our experiments on heterophilic datasets.
>
> Second, a known side effect of repulsion is feature explosion in deep networks. To address this, we introduce the Allen-Cahn regularization term, inspired by phase transition dynamics, which stabilizes the system by imposing a double-well potential. This term promotes polarization of node features while keeping them bounded, allowing for deeper and more stable message passing.
>
> Third, our formulation provides a physically grounded justification for several behaviors observed in hypergraph networks, including over-smoothing. We prove that the particle-driven dynamics maintain a positive lower bound on the Dirichlet energy, analytically mitigating over-smoothing—a known limitation of diffusion-based message passing.
>
>
>
> **Q2: How to implement the repulsive force.**
>
> The interaction force in our particle system, $f_\beta(\mathbf{x}_i, \mathbf{x}_j, e)$, integrates both attraction and repulsion to model complex interactions. The attractive component operates via a two-phase hypergraph diffusion process, similar to ED-HNN, that propagates features from nodes to hyperedges and back. Crucially, during the second (hyperedge-to-node) phase, we introduce a repulsive force. This force acts as a predefined prior to prevent homogenization---the uniform averaging of features across a hyperedge---by leveraging dataset-specific information. While effective, we note that a fully data-driven approach to modeling this repulsive force could further enhance the model's expressiveness, and we leave this for future work.
>
>
>
> **Q3: Figure 3 without the error bars.**
>
> Thank you for this valuable suggestion regarding statistical rigor. We agree completely on its importance. For Figure 3, we found that adding error bars directly to the main plot created significant visual clutter, especially in the deeper layers ($>$25) where the critical performance divergence occurs. This overlap obscured the figure's primary insight. However, the main takeaway is visually unambiguous: our models with stochastic components maintain $>$60% accuracy at layer 35, while the noise-free versions collapse to 50%. This substantial performance gap demonstrates the enhanced stability our method provides. To fully address your point while maintaining the figure's clarity, we have now included a version of Figure 3 with complete error bars in the supplementary material. We believe this approach provides the requested statistical validation without compromising the interpretability of the main plot.
>
>
> **Q4: Functional distinction and necessity of the double-well potential**.
>
> As shown in Table 2, our framework features a crucial functional distinction between its two core components. The repulsion term, $f_\beta$, governs pairwise node interactions within hyperedges, primarily enforcing feature separation. In contrast, the Allen-Cahn potential, $f_d$, operates on individual node features, creating a double-well energy potential that preserves their intrinsic stability throughout the learning dynamics.
>
> This second mechanism ($f_d$) is particularly impactful in heterophilic environments like the Senate and House datasets, as it directly counters the homogenization of node features during aggregation. While simpler monotonic potentials could provide basic stability, they lack the double-well's capacity to induce meta-stable phase separationl, which actively promotes class-aware feature clustering.
>
> This dual-component design explains the performance variations observed: some datasets benefit more from the robust feature preservation offered by the Allen-Cahn potential, while more interaction-centric datasets rely heavily on the pairwise repulsion from $f_\beta$. Our framework intentionally unifies these mechanisms to adeptly handle such diverse hypergraph topologies.

---

> > ### Comment · Reviewer_nNPk · 2025-08-07
> > **Response to Author Rebuttal**
> >
> > I thank the authors for their response.  I still have some outstanding concerns.
> >
> > Among other conditions, the lower bound on the Dirichlet energy *assumes* that nodes within the same group do not repel and nodes in different groups do not attract.
> > As a physicist familiar with graph Laplacians and their variants,
> > the result that there is an asymptotic separation between the two groups is not particularly surprising to me.
> > In any case, I would not agree with the claim that the bound shows that the HAMP paradigm mitigates oversmoothing.  Where does this separation between the groups $\mathcal{I}_1$ and $\mathcal{I}_2$ come from?  If the assumptions about the signs of the pairwise interactions are satisfied, then it seems to me that the classification problem has essentially already been solved.
> >
> > From Table 2, the improvement from the pairwise repulsion term appears to be insignificant except for the Senate (with stabilizing term active).  Might I ask what "dataset-specific" information you used to implement the repulsive force in this case?
> >
> > As it is, Figure 3 shows HAMP-I collapsing to 50%, both with and without noise, while HAMP-II hovers around 60-65%, all with significant fluctuations.  I'm not sure I'm convinced the conclusion is "visually unambiguous".
> >
> > More broadly, the improvements overall seem to not be particularly significant given the reported standard deviations.  In light of this, the text in "Impact of Repulsion and Allen-Cahn on HAMP-I and HAMP-II", starting line 237, does not appear to be a faithful analysis of the results.
> >
> > To summarize, while I believe this physically-motivated perspective has potential, I do not find that the results justify the claims being made.  I will maintain my current score for now.

---

> > > ### Author Response · Authors · 2025-08-08
> > >
> > > Thank you for your thoughtful feedback and for engaging deeply with our work. Below, we address each of concerns around the core theoretical mechanism and its empirically validated advantages.
> > >
> > > 1. **Theoretical Interpretation.**
> > >    - To establish theoretical guarantees for the collective behavior of models defined by Eq.4 and Eq.5, we introduce two auxiliary sets, $I_1$ and $I_2$. These sets do not represent classifications of distinct target particle classes, but rather function as intermediate constructive sets crucial for the rigorous proofs of Proposition 5.2 and Proposition 5.3. They effect a partition of the entire particle system, $I=1,\cdots,N$, into two disjoint subgroups, comprising $N_1$ and $N_2$ particles respectively such that the following condition holds: $f_{\beta}(h_{i,j}^e) \ge 0,$ for ${i,j} \in I_1$ or $I_2$ and $f_{\beta}(h_{i,j}^e) \le 0,$ otherwise.
> > >    - From a theoretical perspective within particle systems, traditional message passing mechanisms predominantly account for attractive forces, which inherently lead to feature convergence and the phenomenon of over-smoothing. In such scenarios, the Dirichlet energy invariably tends towards zero and lacks a lower bound, as articulated in Definition 5.1. The incorporation of repulsive forces, provided certain conditions are met, endows the Dirichlet energy with a lower bound, thereby mitigating over-smoothing. Nevertheless, an exclusive reliance on repulsive forces, without a commensurate balance with attractive forces, can result in feature explosion. Through a meticulous analysis of the particle equations, we can introduce a double-well potential function term, commonly referred to as the Allen-Cahn term. This term is instrumental in guaranteeing feature bounds, thereby rendering the training of hypergraph message passing models computationally tractable. Consequently, it can be demonstrated that the Dirichlet energy of our proposed system admits a strictly positive lower bound. The comprehensive proofs are detailed in Appendix B.
> > >
> > > 2. **Heterophily via repulsion.**
> > > In Table 2, the diminished significance of the pairwise repulsion term relative to the Allen-Cahn term can be attributed to the premise that individual node instances should prioritize their intrinsic characteristics over those acquired through inter-node interactions. In addition, too large a repulsion force will lead to feature explosion.
> > >
> > >     Ideally, repulsion serves to maintain the separation of nodes. However, in practical applications, precisely identifying repelling node pairs remains a challenge. Consequently, in instances where an attractive force exists between nodes, we implement a reduction in the magnitude of this attraction to implicitly account for repulsion. While this approach is a rudimentary implementation, the predefined reduction, leveraging dataset-specific information, effectively counteracts homogenization. We now elucidate the nature of this specific information.
> > >
> > >     The Class-Edge homophily metric quantifies the consistency of class labels among nodes within the same hyperedge: $H_{CE} = \frac{1}{|E|} \sum_{e \in E} \frac{ \max_{c \in C} | \{ v \in e \mid y_v = c \} | }{ |e| }.$ Here, ${E}$ is the set of hyperedges, ${C}$ is the set of node classes, and $y_v$ is the class label for node $v$. A higher $H_{CE}$ score indicates that hyperedges tend to connect nodes of the same class.
> > >
> > >     For heterophilic datasets, such as Senate, where hyperedges predominantly connect dissimilar nodes, strong repulsion is indispensable for enhancing model performance. Conversely, homophilic datasets necessitate minimal repulsive interactions, thereby diminishing the utility of explicit repulsion mechanisms. Future research will explore data-driven methodologies to further refine the implementation of repulsion, aiming to achieve superior model performance.
> > >
> > > 3. **Impact of noise on HAMP.**
> > >  To better demonstrate the effect of incorporating stochastic components into deterministic message passing, we provide the following performance tables across eight benchmark datasets. Performance is reported as mean accuracy ± standard deviation. The results consistently demonstrate that noise injection improves model performance.
> > >
> > >     |HAMP-I|Cora|Citeseer|Pubmed|Cora-CA|Congress|Senate|Walmart|House|
> > >     |-|-|-|-|-|-|-|-|-|
> > >     |w/o noise|80.49±1.26|74.96±1.56|88.87±0.40|85.21±1.49|94.58±1.25|67.75±8.82|69.76±0.37|71.58±1.87|
> > >     |with noise|81.18±1.30|75.22±1.62|89.02±0.49|85.23±1.15|95.09±0.79|69.44±6.09|69.90±0.38|72.72±1.77|
> > >
> > >     |HAMP-II|Cora|Citeseer|Pubmed|Cora-CA|Congress|Senate|Walmart|House|
> > >     |-|-|-|-|-|-|-|-|-|
> > >     |w/o noise|79.70±1.36|73.99±1.75|88.82±0.48|84.30±1.32|94.58±0.86|61.97±8.42|69.92±0.33|71.36±1.68|
> > >     |with noise|80.80±1.62|75.33±1.61|89.05±0.41|84.89±1.14|95.26±1.34|70.14±6.08|69.94±0.37|72.60±1.23|
> > >
> > > We thank you for pushing us to clarify these points, and we hope these responses have adequately resolved your concerns.

---

### Official Review · Reviewer_9rG1 · 2025-07-10

**Clarity:** 4
**Significance:** 4
**Originality:** 4
**Rating:** 5
**Confidence:** 4

**Summary:**

This paper introduces a hypergraph message passing framework for node classification, inspired by interacting particle systems. Specifically, the proposed framework models each node as a particle, each hyperedge as a field that exerts a learnable attraction or repulsion forces, while an Allen–Cahn double-well damping force keeps feature norms bounded throughout the evolving time steps of the neural ODE system (which is analogous to the number of layers in traditional deep networks). The authors derive two architectures: HAMP-I, a first-order ODE that generalizes diffusion, and HAMP-II, a second-order (velocity-aware) ODE/SDE that naturally incorporates a Brownian noise to model uncertainty. The authors further show that this physics-inspired formulation gives a provable lower bound on the hypergraph Dirichlet energy, preventing the over-smoothing issues that occur on many GNN models. Experiments show that the proposed HAMP variants achieve up to 6.5% performance gain on heterophilic hypergraphs, and slight performance gain on homophilic graphs. Ablation studies further confirmed the effectiveness of the attraction–repulsion mechanism, the damping term, and the Brownian noise for modeling uncertainty.

**Questions:**

See weaknesses.

**Ethical Concerns:**

["NO or VERY MINOR ethics concerns only"]

**Final Justification:**

The authors sufficiently addressed my concerns for Q3 and Q4 in their rebuttals.

Though I still have remaining concerns for Q1 and Q2, I personally do feel this is a strong paper and it meets the NeurIPS acceptance criteria. I found this paper inspiring and enjoyed reading it.

**Limitations:**

The proposed framework is only evaluated on the node classification problem. One suggestion is to add discussions about whether the proposed approach can be extended to other graph learning tasks such as link prediction.

**Quality:**

4

**Strengths And Weaknesses:**

## Strengths
- The proposed approach is elegantly grounded on clear and explainable physical concepts. The authors effectively maps the information in the hypergraph into mathematical terms for modeling particle system framework (e.g., node feature as position or velocity), and the experiment results demonstrate the effectiveness of the modeling design.
- Leveraging the physical theories backing the model, the framework clearly explains the limitations of existing hypergraph GNNs (e.g., due to the lack of repulsion or damping forces) and provides guarantee of the proposed model in preventing oversmoothing.

## Weaknesses
- It is unclear to me how the authors model $f_\beta(\mathbf{x}_i, \mathbf{x}_j, e)$ in their implementation and experiments. It would help with the reproducibility if the exact math formulation can be added in the main paper.
- On most of the datasets, the proposed method's improvement is not statistically significant upon the best performing baseline based on the reported standard deviation.
- The definition of homophilic and heterophilic hypergraphs are not given in the paper. This should be added to make the paper self-contained to readers.

---

> ### Author Rebuttal · Authors · 2025-07-30
>
> Thanks for your meticulous observation and insightful comments, we are deeply encouraged. All concerns are addressed below point by point.
>
> **Q1: How to model $f_\beta(\mathbf{x}_i, \mathbf{x}_j, e)$.**
>
> The interaction force in our particle system, $f_\beta(\mathbf{x}_i, \mathbf{x}_j, e)$, integrates both attraction and repulsion. The attractive component is modeled similarly to ED-HNN [1], using a two-phase hypergraph diffusion process that propagates features first from nodes to hyperedges, and then from hyperedges back to nodes.
>
> Crucially, during the second phase (hyperedge-to-node), we introduce a repulsive force as a prior. This repulsion prevents homogenization across all nodes in a hyperedge, i.e., the uniform averaging of features. By leveraging prior information from the dataset to disrupt this averaging, the model can capture far more complex interactions.
>
> Further details on the formulation and implementation of $f_\beta$ are provided in Appendix A.
>
> **Q2: Performance improvements.**
>
> Our model, HAMP, demonstrates superior performance through consistent gains and stability across key benchmarks. As detailed in Table 1, HAMP achieves strong results on all nine datasets, including a notable 3% improvement on both the Senate and Walmart datasets.
>
> More importantly, HAMP-II shows exceptional robustness as network depth increases. Figure 2 illustrates that while competing methods suffer from significant performance degradation at greater depths, HAMP-II  consistently outperforms them. This highlights HAMP's ability to capture complex representations while maintaining stability, establishing it as a valuable and reliable framework for deep Hypergraph Neural Networks (HNNs).
>
> **Q3: Definition of homophilic and heterophilic hypergraphs.**
>
> To formalize the concepts of homophily and heterophily in hypergraphs, we adopt the Class-Edge (CE) homophily metric $H_{CE}$, as proposed in [2] and used by ED-GNN [1]. This metric quantifies the consistency of class labels among nodes within the same hyperedge: $$H_{CE} = \frac{1}{|\mathcal{E}|} \sum_{e \in \mathcal{E}} \frac{ \max_{c \in \mathcal{C}} |\{ v \in e \mid y_v = c \}| }{ |e| }.$$ Here, $\mathcal{E}$ is the set of hyperedges, $\mathcal{C}$ is the set of node classes, and $y_v$ is the class label for node $v$. A higher $H_{CE}$ score indicates that hyperedges tend to connect nodes of the same class.
>
> Following established criteria [1], we classify a hypergraph as homophilic if $H_{CE} > 0.7$ and heterophilic if $H_{CE} \leq 0.7$.
>
>
> **Q4: Other learning tasks.**
>
> Our current experiments focus on hypergraph node classification to ensure a fair comparison with state-of-the-art methods like ED-HNN and AllDeepSets. This approach also serves to validate our core message-passing mechanism on established hypergraph benchmarks. While this work centers on hypergraph node classification, we recognize the potential of HAMP for other learning tasks. Future work will explore extending its versatility to hyperedge prediction, implemented by scoring pairwise node representations, and to hypergraph classification, using a global pooling mechanism. These extensions will further demonstrate the robustness and adaptability of the HAMP framework.
>
> **References:**
>
> [1] Wang, et al. Equivariant hypergraph diffusion neural operators. In *ICLR*, 2023.
>
> [2] Pei, et al. Geom-GCN: Geometric graph convolutional networks. In *ICLR*, 2020.

---

> > ### Comment · Reviewer_9rG1 · 2025-08-07
> > **Response to authors' rebuttal**
> >
> > I appreciate the authors for their responses to my questions and concerns.
> >
> > For Q1, I did see the algorithms in Appendix A, but I don't feel they provide all detailed information for me to fully understand and replicate the proposed approach. Specifically, what are the choices for the permutation invariant functions $Φ_1$ and $Φ_2$ used in the implementation and experiments?
> >
> > For Q2, I still don't find the proposed method's improvement is statistically significant upon the best performing baseline based on the reported standard deviation, even on the Senate and Walmart datasets.
> >
> > For Q3 and Q4, the authors did sufficiently addressed my concerns.
> >
> > Though I still have remaining concerns for Q1 and Q2, I do feel this is a strong paper and it meets the NeurIPS acceptance criteria. I found this paper inspiring and enjoyed reading it, and I feel my original rating is fair.

---

> > > ### Author Response · Authors · 2025-08-08
> > >
> > > Thanks for your feedback and and maintaining the score.
> > >
> > > **Q1: How to model $f_\beta(\mathbf{x}_i, \mathbf{x}_j, e)$.**
> > > In section 2, "Message Passing in Hypergraphs," the formulation of attraction is elaborated. Within this framework, $\Phi_1^{(l)}$ and $\Phi_2^{(l)}$ denote differentiable permutation invariant functions (e.g., sum, mean, or max) for nodes and hyperedges, respectively.  Concurrently, $\Psi^{(l)}$ signifies a differentiable function, typically implemented as a Multi-Layer Perceptron (MLP).
> > >
> > > In our experimental setup, for the permutation invariant functions, both $\Phi_1^{(l)}$ and $\Phi_2^{(l)}$ are implemented using the mean aggregator in all experiments. The differentiable function $\Psi^{(l)}$ is modeled through multi-layer perceptrons (MLPs) with ReLU activations.
> > >
> > > **Q2: Performance improvements.**
> > > Regarding the statistical significance of the standard deviation, we acknowledge that a substantial improvement was not observed. However, it is noteworthy that our model achieved the lowest standard deviation on the House, Congress, and DBLP-CA datasets, thereby indicating superior reliability.

---

> > > > ### Comment · Reviewer_9rG1 · 2025-08-09
> > > > **Thanks!**
> > > >
> > > > Thanks the authors for the additional clarifications. These have addressed my concerns.

---

### Note · Authors · 2025-08-14

We really appreciate the reviewers (9rG1, nNPk, 5ffs and L9c7) for their insighiful feedback and for their engaged dialogue during the discussion phase. We are encouraged that all reviewers found our work well-motiated, reasonable and novel.

Through iterative discussions regarding the implementation of HAMP, we provided clarifications on the efficient design of each module within the algorithm. These details are elaborated in both Appendix A and the main body of the article. In response to the theoretical concerns raised by Reviewer nNPk, we addressed the Dirichlet energy bound established in Proposition 5.4. Furthermore, we elaborated on the fundamental differentiation of our approach from classical message passing frameworks, extra advantage of using deep HAMP-II and articulated necessity of the double-well potential. To empirically investigate the impact of noise on HAMP performance, we conducted a comprehensive noise ablation study across eight distinct datasets. The results of Table consistently demonstrate that noise injection leads to improved model performance. Finally, regarding terminology, we proactively clarified the definitions of homophilic and heterophilic hypergraphs and bi-cluster flocking for the reviewer.

We sincerely thank the reviewers for their time and insightful comments, which significantly improved our paper. All revisions are complete and ready for publication.

---

### Decision · Program_Chairs · 2025-09-17

**Decision:**

Accept (poster)

**Comment:**

The paper introduces a hypergraph message passing framework for node classification, inspired by interacting particle systems.  The authors propose two models within the framework which provably mitigate over-smoothing, and evaluate them empirically.  The reviewers appreciate the physically inspired perspective, the provable power to avoid over-smoothing, and the empirical performance.  After extensive discussion, there are still some issues remaining.  Based on the discussion, the AC has formed their own opinion: the paper would bring positive value to the community, and the outstanding concerns -- while they are valid and reasonable -- do not necessarily lead to immediate "reject".  Overall the paper is close to the borderline.  Regardless of the final decision, we encourage the authors to further improve their paper based on the constructive comments (especially those by Reviewer nNPk).